# Enhanced Climate Reproducibility Testing with False Discovery Rate Correction

Michael E. Kelleher[1] and Salil Mahajan[1]

[1]Computational Hydrology and Atmospheric Sciences Group, Oak Ridge National Laboratory, 1 Bethel Valley Rd, Oak Ridge TN, USA*

**Correspondence:** Michael E. Kelleher (kelleherme@ornl.gov)

**Abstract.** Simulating the Earth's climate is an important and complex problem, thus climate models are similarly complex, comprised of millions of lines of code. In order to appropriately utilize the latest computational and software infrastructure advancements in Earth system models running on modern hybrid computing architectures to improve their performance, precision, accuracy, or all three; it is important to ensure that model simulations are repeatable and robust. This introduces the need for establishing statistical or non-bit-for-bit reproducibility, since bit-for-bit reproducibility may not always be achievable. Here, we propose a short-simulation ensemble-based test for an atmosphere model to evaluate the null hypothesis that modified model results are statistically equivalent to that of the original model. We implement this test in version 2 of the US Department of Energy's Energy Exascale Earth System Model (E3SM). The test evaluates a standard set of output variables across the two simulation ensembles and uses a false discovery rate correction to account for multiple testing. The false positive rates of the test are examined using re-sampling techniques on large simulation ensembles and are found to be lower than the currently implemented bootstrapping-based testing approach in E3SM. We also evaluate the statistical power of the test using perturbed simulation ensemble suites, each with a progressively larger magnitude of change to a tuning parameter. The new test is generally found to exhibit more statistical power than the current approach, being able to detect smaller changes in parameter values with higher confidence.

## 1 Introduction

Thousands of scientists and engineers work tirelessly in efforts to better understand and model the Earth's changing climate. A large portion of this effort has come from the development of Earth system models at modeling centers around the globe, which seek to simulate the atmosphere, ocean, cryosphere, land surface, and chemistry, among other components of the Earth system. These models are comprised of many millions of lines of code and are enormously complex projects worked on by many individuals, so the need arises to verify that contributions to the model code do not have unintended effects on answers

*This manuscript has been authored by UT-Battelle, LLC under Contract No. DE-AC05-00OR22725 with the U.S. Department of Energy. The United States Government retains and the publisher, by accepting the article for publication, acknowledges that the United States Government retains a non-exclusive, paid-up, irrevocable, worldwide license to publish or reproduce the published form of this manuscript, or allow others to do so, for United States Government purposes. The Department of Energy will provide public access to these results of federally sponsored research in accordance with the DOE Public Access Plan (http://energy.gov/downloads/doe-public-access-plan).

produced. Thorough testing of the output from these models is routinely conducted, assessing if the results are identical (bit-for-bit) or not. If the results are not bit-for-bit identical, statistical checks are also conducted to ensure the simulated climate of the model or component has not significantly changed, unless that is the intended effect. A variety of methods are available (Milroy et al., 2018; Wan et al., 2017; Baker et al., 2015; Mahajan et al., 2017, 2019b), each of which performs a statistical comparison between a reference ensemble and test simulation or ensembles. Here, an ensemble indicates a set of model runs each initialized with similar but slightly perturbed initial conditions, generally only at machine-precision levels.

The Energy Exascale Earth System Model (E3SM, Golaz et al. (2022); E3SM Project (2023)) uses a suite of tests which run the model under a variety of configurations and methods using the Common Infrastructure for Modeling the Earth (CIME) software to setup, build, run, and analyze the model. The tests are run at varying frequencies from nightly to weekly, testing both the latest science and performance updates and maintenance branches, which only receive compatibility updates and should be bit-for-bit identical. A subset of these tests are statistical reproducibility tests (non-bit-for-bit tests), which determine through a comparison of control and perturbed ensembles, whether or not the simulated climate has changed as a result of modifications to the model code or infrastructure. These are the Time Step Convergence test (TSC, Wan et al. (2017)), the Perturbation Growth New test (PGN, related to the work in Rosinski and Williamson (1997)), the multi-testing Kolmogorov-Smirnov test (MVK, Mahajan et al. (2019b)), and the MVK-Ocean test (MVK-O, Mahajan (2021)). The first three test the reproducibility of the E3SM Atmosphere Model, (EAM), and the last tests the Model for Prediction Across Scales-Ocean (MPAS-O), the ocean component in E3SM.

The TSC test evaluates numerical convergence by comparing ensemble differences at two small time step sizes (i.e., 1 sec and 2 sec) using a Student's $t$-test on root mean squared difference (RMSD) values, under the assumption that numerical solutions should converge as the time step decreases (Wan et al., 2017). The PGN test, in contrast, assesses stability by comparing the state of the atmosphere after one time step across perturbed ensemble members, identifying whether small initial differences grow inconsistently through individual physics parameterizations. Unlike the shorter duration TSC and PGN tests, the MVK and MVK-O tests use year-long and two-year long simulation ensembles respectively, allowing them to assess the cumulative impact of code or configuration changes on the model's climatology after internal variability has saturated, providing a more robust evaluation of long-term climate statistics.

MVK evaluates the null hypothesis that a modified model simulation ensemble is statistically equivalent to a baseline ensemble. It applies a two-sample Kolmogorov-Smirnov test to over 100 output variables and counts how many show statistically significant differences between the two ensembles. If this count exceeds a critical value threshold as expected from internal variability and derived via bootstrapping, the two simulations are considered to have different simulated climates (Mahajan et al., 2017). Further details on the MVK are provided in Section 2.1.

Here, we propose a new testing approach that improves on the MVK and provides a two-fold benefit over it. Firstly, from a human usability perspective, it is desirable for the test to reduce the number of false positives (Type I error rates; where two simulations are erroneously labeled as statistically different), without affecting the false negative rates (Type II error rates; where two simulation ensembles are erroneously labeled as statistically similar). Operationally, MVK has been exhibiting a false positive rate of about 7.5%, despite the prescribed significance level of 5%, since its induction into the test suite (as

discussed in Section 4.7). Other previous works suggest that implementing a false discovery rate (FDR) correction (by adjusting significance thresholds based on the number and rank of p-values, discussed in more detail in Section 2.3), reduces the false positive rate when multi-testing as compared to using bootstrapping-derived critical value threshold for hypothesis testing (Ventura et al., 2004; Wilks, 2006, 2016; Mahajan, 2021). This is the methodology used by MVK. Some frameworks, including those presented in Mahajan (2021); Zeman and Schär (2022), seek to reduce false positive rates, and do so successfully, but at a greater computational cost, as they work with grid-box level data. The work in Zeman and Schär (2022) compares larger sub-ensemble sizes which reduces the false positive rate lower than an FDR correction, but adds to the computational cost, while Mahajan (2021) uses the same ensemble sizes, but applies an FDR correction at each grid box in the ocean model. The new testing approach for the atmosphere model thus implements an FDR correction when evaluating multiple variables in the atmosphere model output. Secondly, it will be useful to reduce the computational cost and time of deriving the critical value thresholds for the MVK. The bootstrapping procedure is time-consuming and computationally expensive because of its need for large control ensembles. And, it would need to be conducted again after a significant enough departure from the original model code to establish a new critical value threshold. Substantial model code changes to numerics and physics can alter internal variability and shift the distributional properties of output fields. FDR correction theoretically asserts the critical value threshold for global null hypothesis evaluation (as discussed in Section 2.3) is 1 or more rejected field, thus eliminating the need for conducting a large control ensemble simulation to determine this threshold. A large control ensemble is required, though, to demonstrate that the theoretical critical value is equivalent or better than the one arrived at by bootstrap sampling of the large ensemble.

This paper thus seeks to answer two questions: can a testing framework using FDR correction reduce the number of false positives in an operational setting as compared to the MVK without impacting its statistical power (false negative rates) and can it eliminate the need for extensive and expensive analysis of large ensembles each time the model is updated significantly? We also explore the impact of applying the BH-FDR correction to different underlying statistical test used to evaluate the differences in distribution, using the Mann-Whitney U and Cramér-von Mises tests in addition to the K-S test.

The following section discusses the MVK test and its pitfalls in more detail and describes the FDR approach as implemented here. Section 3 lists the simulation ensembles conducted to evaluate both the MVK and the new testing framework. Section 4 discusses our results on the evaluation of the false positive and negative rates of these testing frameworks. Finally, our results are summarized in Section 5 with a brief discussion of caveats of the study and future direction.

## 2 Methods

### 2.1 Multi-testing Kolmogorov-Smirnov Test

The multi-testing Kolmogorov-Smirnov (MVK) test, as implemented in the CIME for the atmosphere model of E3SM, compares two independent $N = 30$ member ensembles. It is used in nightly testing, comparing a baseline ensemble, generated after each approved "climate changing" code modification, and a test ensemble, newly generated each day. The baseline ensemble is generated following approvals from domain scientists who have expertise related to the newly introduced code. The

E3SM model is run at "ultra-low" resolution, (called ne4pg2, $\approx 7.5°$ atmosphere), for 14 months. The first two months are discarded as the system reaches quasi-equilibrium. Annual global means of each of the 120 standard output fields of the E3SM Atmosphere Model (EAM) are then computed for each ensemble member. Then, for each field, the null hypothesis ($H_1$, also referred to as the local null hypothesis here) is evaluated. The local null hypothesis asserts that the sample distribution function of the annual global mean of that field estimated from the baseline ensemble (from $N$ data points) is statistically similar to that of the new ensemble is evaluated. The two sample Kolmogorov-Smirnov (K-S) test at a significance level of $\alpha = 0.05$ is used for testing $H_1$. The K-S test is a non-parametric statistical test to compare cumulative or empirical distribution functions. The larger null hypothesis ($H_0$, also referred to as the global null hypothesis here) that the two ensembles have identical simulated climates is then evaluated for a significance level of $\alpha$. The test statistic, $t$, for testing this larger null hypothesis is defined as the number of fields that reject $H_1$. $H_0$ is rejected if the number of fields rejecting $H_1$ is greater than a critical value threshold (found to be 13, Mahajan et al. (2017)), and the test issues a "fail". The null distribution of $t$, and hence the critical value threshold at a significance level of $\alpha$, was empirically derived by randomly sampling two $N$-member ensembles from a 150-member control ensemble of an earlier version of E3SM, and computing $t$, 500 times (Mahajan et al., 2017). While regional means, local diagnostics, and extremes can provide valuable perspectives on ensemble behavior, here we focus on annual global means, following previous work (Baker et al., 2015; Mahajan et al., 2017). Atmospheric variability is highly heterogeneous in both space and time, with sharp gradients and transient fluctuations that introduce substantial sampling noise into regional ensemble diagnostics. Global averaging suppresses stochastic, weather-driven variability and reduces dimensionality, enabling systematic differences between two ensembles to be more clearly identified. Further, extremes require much larger ensembles to be reliably assessed, as Mahajan et al. (2017) demonstrated. With ensembles of about sixty members, extremes of temperature and precipitation were often statistically indistinguishable even when mean distributions diverged, underscoring their limited sensitivity in this context.

### 2.1.1 Potential pitfalls of multi-testing K-S Test

MVK requires a large control ensemble in order to capture the variability of the model and establish proper critical value thresholds for the number of rejected local null hypotheses before rejection of the global null hypothesis is considered (Wilks, 2006). It also, as discussed in Wilks (2006), does not put weight onto fields rejected very strongly. That is, those fields with extremely small p-values. If a few fields are rejected with near certainty, this likely indicates global significance. But, if the total number of these fields does not exceed the predetermined threshold, the overall result is global null hypothesis acceptance, which results in a lower power to detect differences. Finally, in practice this approach has a larger Type I (false positive) error (see Sections 4.2, 4.7; Table 2) than desired. While possible to decrease the significance level $\alpha$ to reduce the false positive rates, this in turn decreases the statistical power of the test to detect small changes which is not desirable.

### 2.2 Additional statistical tests

The Kolmogorov-Smirnov test used in the MVK framework is just one non-parametric test available to ascertain differences between distributions of random variables. To add robustness to the assessment of distributional differences, the Mann-Whitney

U (also called the Wilcoxon rank sum) and Cramér-von Mises tests were also performed to compare each set of ensembles. The Mann-Whitney U test (M-W hereafter), ranks all samples from the two groups, and compares the sum of those ranks between the two groups (Mann and Whitney, 1947). The Cramér-von Mises test (C-VM hereafter) used here compares two empirical distributions using the quadratic distance between them (Anderson, 1962). These tests present an advantage in statistical power over the K-S test, as more weight is given to the distribution tails (Gentle, 2003), resulting in higher sensitivity to extreme values, though at the cost of being more sensitive to outliers. Both additional tests are also non-parametric tests, meaning that limited assumptions are asserted about the shape of the data.

## 2.3 False Discovery Rate Correction

When conducting simultaneous multiple null hypothesis tests as in the MVK above, Type I error rate inflation occurs, leading to a higher overall probability of false positives, since the number of expected null hypothesis rejections increases with each additional test (Benjamini and Hochberg, 1995). This means that increased false positives can occur when testing many p-values at once. While a significance level (e.g., $\alpha = 0.05$) controls the probability of a false positive for a single test, conducting many tests increases the chance that at least one result will appear significant purely by chance even if all the null hypotheses are true. This implies that the overall proportion of false discoveries can become unacceptably high, undermining the reliability of the findings. MVK accounts for multi-testing by using a re-sampling strategy, which was found to give similar results as compared to permutation testing (Mahajan et al., 2019b). In addition to permutation testing and bootstrapping approaches, other approaches for correcting Type I error rate inflation associated with multi-testing include family wise error rate correction (e.g. Bonferroni correction), and the false discovery rate (FDR) (Wilks, 2006; Ventura et al., 2004), the latter of which is used here. The Bonferroni correction adjusts the significance level by dividing the desired significance level, $\alpha$, by the number of tests conducted, but is known to have reduced statistical power (Wilks, 2006; Ventura et al., 2004). Here, we use the Benjamini-Hochberg (BH) FDR correction approach (Benjamini and Hochberg, 1995). The BH-FDR approach has been shown to effectively control for Type I error rate inflation while also exhibiting more power than other approaches (Ventura et al., 2004; Wilks, 2006). It is widely used in Earth system studies for spatial analysis (Wilks, 2006; Renard et al., 2008; Whan and Zwiers, 2017) and has also been applied to solution reproducibility testing of ocean models (Mahajan, 2021). Other methods of false discovery rate correction (Benjamini and Yekutieli (2001), Bonferroni adjustment) were also examined and found to have lower power than the BH-FDR method used here. These corrections remove more rejections than the BH-FDR correction and result in fewer global null hypothesis rejections, so they are less sensitive to small parameter changes. While the original BH-FDR (Benjamini and Hochberg, 1995) description asserted independence, it has been found in more recent work (e.g. Benjamini and Yekutieli (2001); Ventura et al. (2004)) that the independence of the p-values is not a strict requirement.

Similar to MVK, we use the two sample K-S, the two sample C-VM, and the two sample M-W tests to evaluate the local null hypothesis ($H_1$) for each field that their sample distribution functions are statistically identical across the two ensembles. Also, similar to the MVK, the global null hypothesis ($H_0$) being tested is that the two ensembles are statistically similar. We use the `statsmodels` (Seabold and Perktold, 2010) Python package for applying BH to reproducibility testing. This corrects the critical threshold, $\alpha$, for evaluating local null hypothesis ($H_1$) rejection with the $k^{th}$ sorted p-value, where $k$ is defined in

equation 1 (see Eq. 2 of Ventura et al. (2004) and Eq. 3 of Wilks (2016)).

$$k = \max_{i=1...m} \left[ i : p_{(i)} \leq q^* \frac{i}{m} \right] \qquad (1)$$

Where $p_{(i)}$ is the $i^{th}$ smallest p-value, $m$ is the total number of p-values (equivalent to the number of hypothesis tests), and $q^*$ is the chosen limit on the false discovery rate, typically chosen to be $q^* = \alpha$. This $q^*$ is defined as the upper limit of the false positive rate using the FDR-BH methods, and is chosen at the 5% level as it balances false negative and false positive rates. A decrease in $q^*$ would decrease false positives, but it would be offset by an increase in false negative errors. The result of Eq. 1, $k$, is thus the index of the largest p-value which is less than $q^*i/m$, and all p-values with an index less than k (e.g. $p_{(1)} \dots p_{(k)}$) are rejected. Now, the global null hypothesis, $H_0$, that can be framed as all $H_1$ are true, is rejected at the global significance level of $\alpha$ if *any* $H_1^{(i)}$ is rejected (Renard et al., 2008; Wilks, 2006; Ventura et al., 2004). Thus, the global null hypothesis is rejected if *any* p-value is rejected, said another way, if $k \geq 1$. In other words, the critical value threshold for the test statistic, $t$, which is the number of fields rejecting $H_1$, is equal to one for $H_0$ for BH-FDR. This theoretical critical value threshold of one, when multi-testing with FDR corrections, has been used widely for field significance testing to address the multiple comparisons problem inherent in spatial analyses in the climate and meteorological studies (Wilks, 2006; Renard et al., 2008; Burrell et al., 2020). More details about the BH-FDR as applied to earth system science can also be found in these studies.

This BH-FDR as applied to reproducibility testing will be deemed useful here if it performs at least as well at detecting altered simulated climates (statistical power) as the MVK while also reducing the number of false positives. We investigate these characteristics for each approach using suites of simulation ensembles with controlled modifications in Section 4.

## 2.4   Illustration of FDR approach with test cases

Figure 1 illustrates the FDR approach using two example simulation ensemble comparisons, described later in Section 3. At left, a control comparison is shown where each ensemble member differs by a machine precision perturbation only (expected to globally pass, but have local rejections), and at right a perturbed ensemble experiment is compared to the control with known solution changes (expected to be globally rejected). Note that several p-values in the control self comparison are below the threshold $\alpha$, but none fall below $\alpha^{FDR}$, leading to an acceptance of the global null hypothesis for this experiment. In the comparison between control and perturbed ensembles there are fewer p-values which fall below the corrected $\alpha^{FDR}$ than fall below the nominal $\alpha$, despite this, the global null hypothesis is still rejected, as at least one field is rejected. This is similar to Fig. 2 from Ventura et al. (2004), which also illustrates the impact of a variable $\alpha$.

## 3   Simulation Ensembles

We conduct a suite of multi-member ensembles to evaluate the false positive and false negative rates of BH-FDR approach. Additionally, having undergone significant updates to software (Golaz et al., 2022), we reassess the critical value threshold of MVK at which two ensembles of E3SMv2.1 are statistically distinguishable (the critical value threshold having been previously computed from a control ensemble of an earlier model version) using this ensemble suite. Each ensemble is generated by

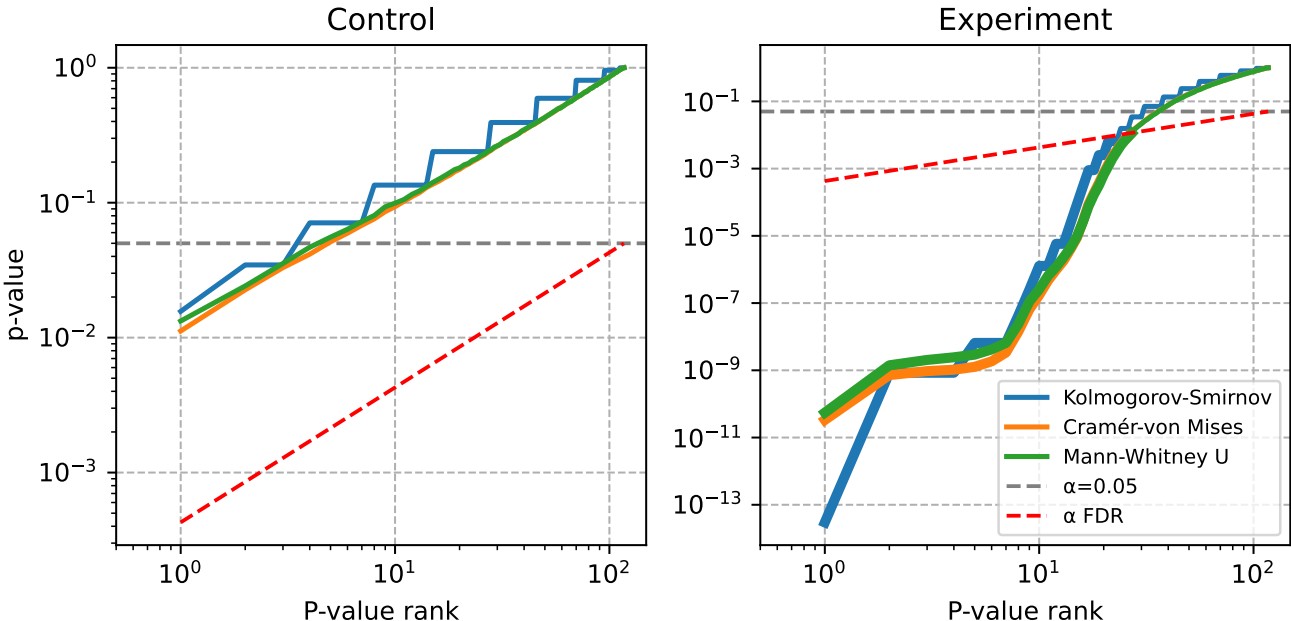

**Figure 1.** Sorted p-values for two selected experiments, shown in blue, orange, and green for the Kolmogorov-Smirnov, Cramér-von Mises, and Mann-Whitney U tests respectively. These lines are thicker where p-values are rejected based on BH-FDR correction. The dashed grey line shows the nominal $\alpha = 0.05$, and the dashed red line shows the FDR corrected $\alpha^{FDR} = q^* i/m$.

varying the initial conditions by near-machine precision perturbations ($O(10^{-10})$) to the temperature field at each grid point for each ensemble member. One of these ensembles is the control ensemble which refers to the unmodified default E3SMv2.1 model with default values of all tuning parameters. Other generated ensembles are differentiated from the control ensemble by

varying the value of a tuning parameter from its default value by different magnitudes (see Table 1 for details). Here, we vary three tuning parameters separately. Version 1 of E3SM was found to be highly sensitive to a parameter termed `clubb_c1`, somewhat sensitive to `zmconv_c0_ocn`, and weakly sensitive to the `effgw_oro` (Qian et al., 2018), similar parametric sensitivity was found in the Community Atmosphere Model version 6 (CAMv6) (Eidhammer et al., 2024). `clubb_c1` is the constant associated with dissipation of variance of $\overline{w'^2}$ (where $w$ is vertical wind speed), `zmconv_c0_ocn` is the deep

convection precipitation efficiency over ocean grid-points, and `effgw_oro` is the gravity wave drag intensity (see Table 1 of Qian et al. (2018)). We choose these three parameters to generate ensembles to capture a range of sensitivities in version 2 of E3SM. As in the operational case described in 2.1, the simulation ensembles are run at "ultra-low" resolution with a 14-month simulation duration. A 120-member ensemble is generated for each tuning parameter change, along with the control ensemble.

| Parameter | % Change | Parameter value |
|---|---|---|
| `effgw_oro` | **0.0** | **0.375** |
| | 1.0 | 0.3788 |
| | 10.0 | 0.4125 |
| | 20.0 | 0.4500 |
| | 30.0 | 0.4875 |
| | 40.0 | 0.5250 |
| | 50.0 | 0.5625 |
| `clubb_c1` | **0.0** | **2.400** |
| | 1.0 | 2.424 |
| | 3.0 | 2.472 |
| | 5.0 | 2.520 |
| | 10.0 | 2.640 |
| `zmconv_c0` | **0.0** | **0.0020** |
| | 0.5 | 0.00201 |
| | 1.0 | 0.00202 |
| | 3.0 | 0.00206 |
| | 5.0 | 0.00210 |

**Table 1.** List of simulations for each tuning parameter

The `effgw_oro` and `clubb_c1` ensembles were run on Argonne National Laboratory's Chrysalis machine and built using the Intel compiler v20.0.4, while the `zmconv_c0_ocn` ensembles were run on US Air Force HPC11 at Oak Ridge National Laboratory's Oak Ridge Leadership Computing Facility (OLCF), compiled using the GNU compilers v13.2. An additional control ensemble was also performed on HPC11.

To further evaluate the BH-FDR approach, we conduct two additional ensemble simulations on Chrysalis to test model sensitivity to compiler optimization choices. The default optimization flag for E3SM is "-O3", and is used to generate the control and other ensembles with tuning parameter changes. The two different optimization test ensembles are titled `opt-O1` and `fastmath`, and are compiled with optimization flags "-O1" and "-O3 -fp-model=fast" respectively. Previous work suggests that using optimization "-O1" is expected to not produce a significantly different simulated climate than the default (Baker et al., 2015; Mahajan et al., 2017). Mahajan et al. (2017) found that using optimization flag "fast" with "Mvect" resulted in a statistically different climate compared to the default, which used an optimization of "-O2" using the PGI compiler.

The "-O1" optimization turns off most of the aggressive optimizations used for the "-O3" level, including loop vectorization, loop unrolling, and global register allocation, while enabling the "-fp-model=fast" fast floating point model allows the compiler to be less strict in its handling of floating point arithmetic (Intel Corporation, 2023). This means using "-O1" in place of "-O3" ought to result in slower operation but similar results, while using "-fp-model=fast" could result in different answers

under specific conditions, including if there are "NaN" or not-a-number values present. Though in the case of E3SM, "-fp-model=fast" is already used in the compilation of several source files, thus adding it as a global option only changes those where it is not in use already.

## 4 Results

### 4.1 Estimating Critical Value Threshold for MVK

E3SMv2.1 has undergone several scientific feature changes as well as software infrastructure changes since the release of E3SMv0 which was used to estimate the critical value threshold for null hypothesis testing using MVK (Golaz et al., 2022). Here, we estimate the null distribution of the test statistic of MVK which is more representative of E3SMv2.1. We use a bootstrapping (re-sampling) strategy to derive the null distribution and the empirical critical value threshold of the test statistic, $t$, which is the number of variables falsely rejecting the true null hypothesis $H_1$ at $\alpha = 0.05$, for the global null hypothesis using for the MVK test for E3SMv2.1. For this analysis, two 30 member ensembles were drawn, without replacement, from the 120 member control ensemble (with $> 10^{18}$ possible ways of drawing a 30 member ensemble). By drawing two ensembles from the same population, this establishes an expected value for how many fields would be rejected by random chance when using two ensembles which have the same simulated climate. $t$ is computed for each such draws of 30-member ensemble pairs and then this procedure is repeated 1000 times. This procedure was then applied to each parameter adjustment ensemble separately, comparing two random draws from each large ensemble, in an effort to expand the sample size to estimate the null distribution. It is also applied to the ensembles with changes to compiler optimizations, thus the empirical threshold for rejecting the global null hypothesis is computed from 2040 ensemble members. The null distribution of $t$ is representative of the internal variability of the model as the drawn ensemble pairs are part of the same population with ensemble members differing only in the initial conditions at machine precision level perturbations. Figure 2 illustrates the null distribution of $t$ with a box and whiskers plot derived from each 120-member ensemble using the K-S test. The critical value threshold is also estimated as the 95th percentile of $t$, which ranges from 10 to 13 for different ensembles. We thus set the critical value threshold for MVK (K-S) at $\alpha = 0.05$ as the median of those values, found to be 11. This procedure was repeated using the other two statistical tests with similar results, though the thresholds for C-VM and M-W were both found to be larger, at 16.

### 4.2 False positive rates: Uncorrected and BH-FDR approach

The bootstrapping method described above in Sec. 4.1 to determine the critical value threshold for global null hypothesis testing for MVK can also be used to estimate the false positive rates. Since ensemble pairs are drawn from the same population, each drawn pair that rejects the null hypothesis is a false positive. The thresholds for global rejection are set at K-S: 11, C-VM: 16, and M-W: 16 for null hypothesis testing at $\alpha = 0.05$. For the BH-FDR approach applied to each statistical test, any field rejecting the null hypothesis after false discovery rate correction implies a rejection of the larger null hypothesis.

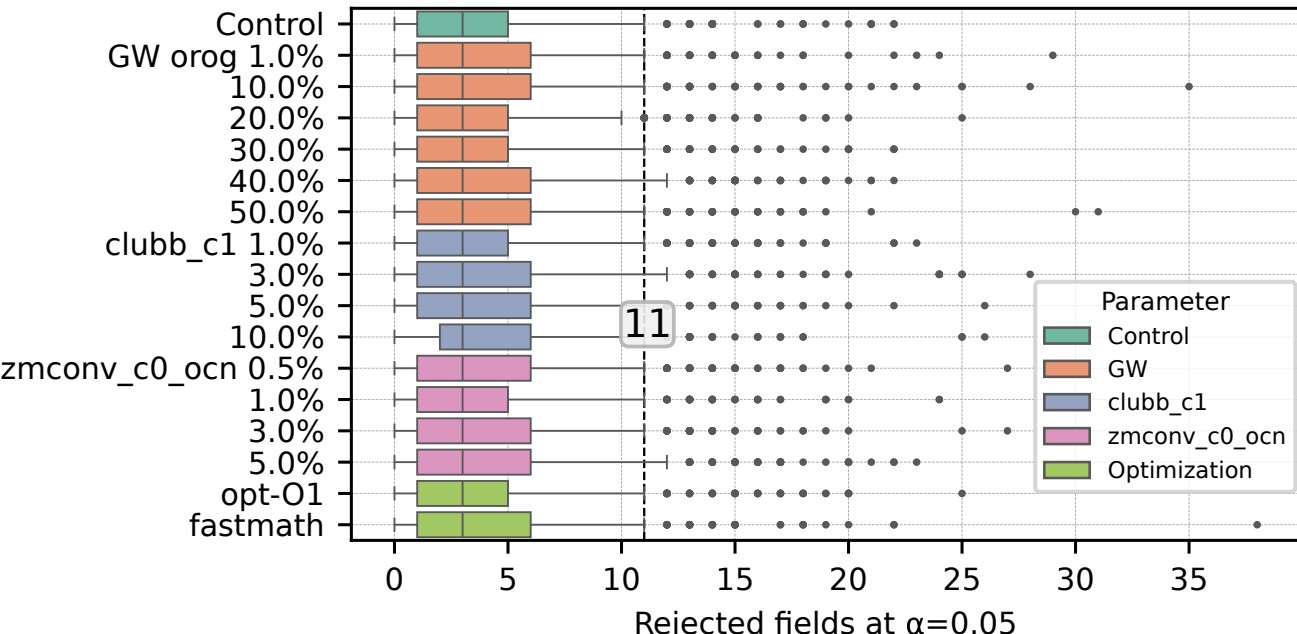

**Figure 2.** Box plot of number of rejected output fields using the K-S test for each ensemble self-comparison. The solid center line of each box represents the median, the box represents the inter-quartile range ($25^{th}$-$75^{th}$ percentiles), the whiskers are plotted at the $5^{th}$-$95^{th}$ percentiles, and outliers marked beyond that range. The dashed vertical line is the median of all 95th percentiles which is 11 fields.

For each of the thirteen 120-member ensembles conducted (control, tuning parameter changes and optimization change ensembles), a 1000 iteration bootstrapping analysis is performed separately, and the false positive rates are computed for each under each statistical test and the BH-FDR testing approach. Table 2 details the false positive rates derived from each analysis. The mean of these 17 values for the MVK is 0.046, which can be expected to be near the prescribed $\alpha$ since the critical value threshold was estimated from the same population (set of ensembles), although with different bootstrap samples, similarly for M-W and C-VM tests, the mean false positive rate are 0.048 and 0.049 respectively. The mean false positive rate under the BH-FDR approach is lower for all tests at 0.032, 0.035, and 0.040 for MVK, M-W, and C-VM respectively. Also, the 95th percentile of false positive rates for the 17 ensemble comparisons are also reduced BH-FDR approach for all statistical tests. The above indicate that the application of the FDR correction works as intended and based on the Lemma of Theorem 1 in Benjamini and Hochberg (1995), the FDR puts an upper limit to the level of false discovery at $q^*$ (which here is chosen as $q^* = \alpha = 0.05$), thus false positive rates lower than $\alpha$ are expected.

### 4.3 False Negative Rates and Statistical Power: MVK and BH-FDR

To evaluate the magnitude of change that the tests can detect confidently, we again rely on bootstrapping following previous work (Mahajan et al., 2019b, a; Mahajan, 2021). For each tuning parameter change, 30 ensemble members each are drawn from

| Statistical Test | Kolmogorov-Smirnov (K-S) | | Mann-Whitney (M-W) | | Cramér-von Mises (C-VM) | |
| Method | Uncorrected | BH-FDR Corrected | Uncorrected | BH-FDR Corrected | Uncorrected | BH-FDR Corrected |
|---|---|---|---|---|---|---|
| **Simulation Ensemble** | | | | | | |
| Control | 0.049 | 0.028 | 0.049 | 0.029 | 0.046 | 0.035 |
| GW orog 1.0% | 0.054 | 0.032 | 0.036 | 0.025 | 0.034 | 0.026 |
| GW orog 10.0% | 0.044 | 0.022 | 0.048 | 0.033 | 0.051 | 0.040 |
| GW orog 20.0% | 0.061 | 0.039 | 0.051 | 0.033 | 0.055 | 0.041 |
| GW orog 30.0% | 0.048 | 0.032 | 0.046 | 0.031 | 0.054 | 0.036 |
| GW orog 40.0% | 0.035 | 0.031 | 0.050 | 0.044 | 0.051 | 0.044 |
| GW orog 50.0% | 0.053 | 0.028 | 0.049 | 0.040 | 0.046 | 0.043 |
| clubb c1 1.0% | 0.041 | 0.049 | 0.040 | 0.029 | 0.043 | 0.029 |
| clubb c1 3.0% | 0.053 | 0.034 | 0.058 | 0.045 | 0.056 | 0.050 |
| clubb c1 5.0% | 0.049 | 0.031 | 0.050 | 0.036 | 0.054 | 0.037 |
| clubb c1 10.0% | 0.044 | 0.029 | 0.046 | 0.032 | 0.048 | 0.036 |
| zmconv c0 ocn 0.5% | 0.041 | 0.029 | 0.049 | 0.042 | 0.050 | 0.047 |
| zmconv c0 ocn 1.0% | 0.045 | 0.046 | 0.063 | 0.038 | 0.051 | 0.050 |
| zmconv c0 ocn 3.0% | 0.032 | 0.028 | 0.045 | 0.030 | 0.047 | 0.040 |
| zmconv c0 ocn 5.0% | 0.040 | 0.040 | 0.041 | 0.034 | 0.046 | 0.039 |
| opt-O1 | 0.060 | 0.032 | 0.053 | 0.036 | 0.058 | 0.040 |
| fastmath | 0.059 | 0.034 | 0.043 | 0.030 | 0.043 | 0.040 |
| **Mean** | 0.048 | 0.033 | 0.048 | 0.035 | 0.049 | 0.040 |
| **95th %tile** | 0.060 | 0.047 | 0.059 | 0.044 | 0.056 | 0.050 |

**Table 2.** False positive rates for self-comparison bootstraps (per 1000 iterations for each ensemble) at $\alpha$=0.05, and the mean and $95^{th}$ percentile false-positive rate over all self-comparison bootstraps.

the control and that tuning parameter change ensemble. The test statistic, $t$, is then computed for each uncorrected statistical test (K-S, M-W, and C-VM) and the BH-FDR corrected version of each, then the six (three uncorrected, three BH-FDR corrected) tests are conducted on the ensemble pair. This procedure is then repeated 1000 times. To illustrate the impact of progressively increasing the tuning parameter on the model climate, Fig. 3 shows the 95th percentile of $t$ from the 1000 hypothesis tests for each tuning parameter change for effgw_oro, clubb_c1, and zmconv_c0_ocn. As the magnitude of the tuning parameter change to the model increases, $t$ increases for both uncorrected and BH-FDR corrected approaches. To reiterate, an increase in $t$ indicates an increase in the number of fields rejecting the local null hypothesis, $H_1$. Fig. 3 also re-iterates the lower sensitivity of E3SM to the orographic gravity wave drag parameter than both the C1 parameter from the CLUBB cloud parameterization scheme, and C0 over ocean in the ZM convection scheme. Smaller percentage changes in clubb_c1 and zmconv_c0_ocn result in large changes to $t$ as compared to effgw_oro, where larger percentage changes are needed for

similar changes in $t$. The K-S test, as expected from the control threshold estimation, rejects the fewest number of fields at each level, while the C-VM and M-W tests reject nearly the same number. However, the number of rejected fields above each test's respective control threshold appears similar.

Fig. 3 also points towards the detectability of modifications to the model by the tests. For the `effgw_oro` and `clubb_c1` parameters, a 1% change in their value does not result in a change in the simulated climate that is easily detectable by the tests

since >95% of the bootstrapped ensemble pairs, $t$ is less than the critical value threshold of the tests (11 for K-S, 16 for M-W and C-VM, and one for BH-FDR). This indicates that that 95% of the 1000 bootstrap comparisons between the control and, for example, the 10% change to `effgw_oro`, have 13 or fewer of the 117 output fields rejected for the K-S test, 19 or fewer fields for the C-VM test, and 20 or fewer for the M-W test, and 1, 2, and 2 of 117 output fields rejected for BH-FDR corrected versions for each. The 95% level is chosen here as it is the inverse of our significance level $\alpha = 0.05$, and as such one could

expect to see 5% of bootstrap iterations failing the test on random chance, thus if fewer than 5% have passed, this is a strong indicator that the two ensembles are significantly different.

Increasing `effgw_oro` to 10%, `clubb_c1` to 3%, and `zmconv_c0_ocn` to 1% results in some of the bootstrap iterations having exceeded or met the critical value threshold for both MVK and BH-FDR. As the magnitude of tuning parameter change increases, the number of bootstrap iterations crossing the critical value threshold also increases.

A formal estimate of the statistical power ($P$, rate of correctly rejecting a false null hypothesis), also representative of the false negative rates ($1 - P$, incorrectly accepting a false null hypothesis), of these tests is illustrated in Fig. 4. It shows the number of bootstrap iterations where the tests correctly reject $H_0$ at a significance level of $\alpha = 0.05$ and is indicative of the likelihood of the tests detecting a modification to the model. $P$ can be computed by dividing the ordinate (y-axis) values by the total number of bootstrap iterations (1000). Similar to Fig. 3, Fig. 4 shows that as the magnitude of a tuning parameter is

increased, the number of bootstrap iterations rejecting $H_0$ also increases. For a 1% change to `effgw_oro` or `clubb_c1` only about 30–40 bootstrap iterations reject $H_0$, which implies that there is only a 30–40 out of a 1000 (3–4%) chance that a change of this magnitude could be detected by the tests. At a 0.5% change to `zmconv_c0_ocn`, about 60–70 bootstrap iterations reject $H_0$, implying a 6–7% chance of detecting a change of this magnitude using these tests. At the $\alpha = 0.05$ significance level it is expected that $\approx 5\%$ of the tests will be rejected by random chance. Increasing `effgw_oro` to 10% results in more than 50

bootstrap iterations (5%) rejecting $H_0$, indicating that it unlikely to be caused by random chance at the 0.05 significance level, it still exhibits a low likelihood of being detected by the tests (about 60 and 80 out of a 1000 chance for MVK and BH-FDR tests respectively). As the magnitude of change to `effgw_oro` is increased, the likelihood of detecting a change by the tests increases. For a change of 40% to `effgw_oro` there is a >90% chance of being detected by both the tests and it reaches nearly a 100% for a larger change. Similarly, for `clubb_c1`, a change of 5% results in greater than 80% (90%) chance of

being detected by the MVK (BH-FDR) test, and nearly a 100% chance of detection for a change of about 10% change to its magnitude. Overall, BH-FDR approach exhibits greater statistical power than MVK for almost all tuning parameter changes, allowing increased confidence in detecting changes, adding to its advantages. This is consistent with previous works (Benjamini and Hochberg, 1995; Wilks, 2006), that suggest that BH-FDR approach generally exhibits greater power than bootstrapping methods. Figure 5 shows the power difference between uncorrected and BH-FDR corrected approaches. For the `clubb_c1`

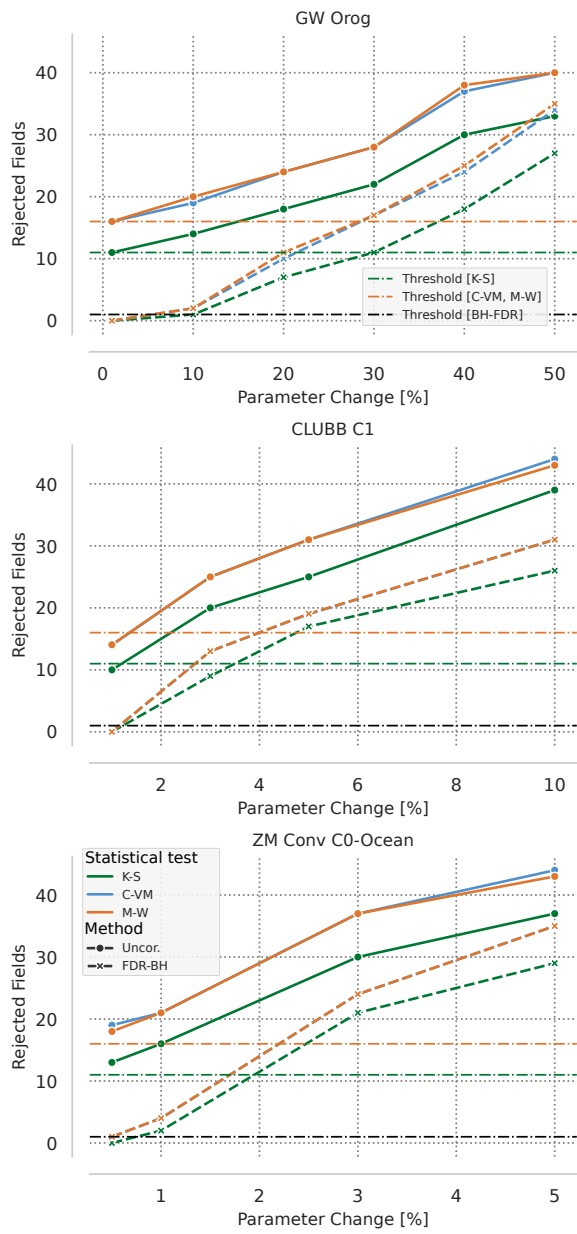

**Figure 3.** 95th percentile of the number of fields with statistically significant differences from the control ensemble (reject the local null hypothesis, $H_1$ at the $\alpha = 0.05$ significance level) on the y-axis by percentage change in tuning parameter along the x-axis. Solid green, blue, and orange lines represent K-S, C-VM, and M-W tests respectively, dashed represents BH-FDR versions of each in the same color. The dashed horizontal lines represent global null hypothesis critical value thresholds for K-S, C-VM, and M-W tests (green, orange respectively) and BH-FDR (black).

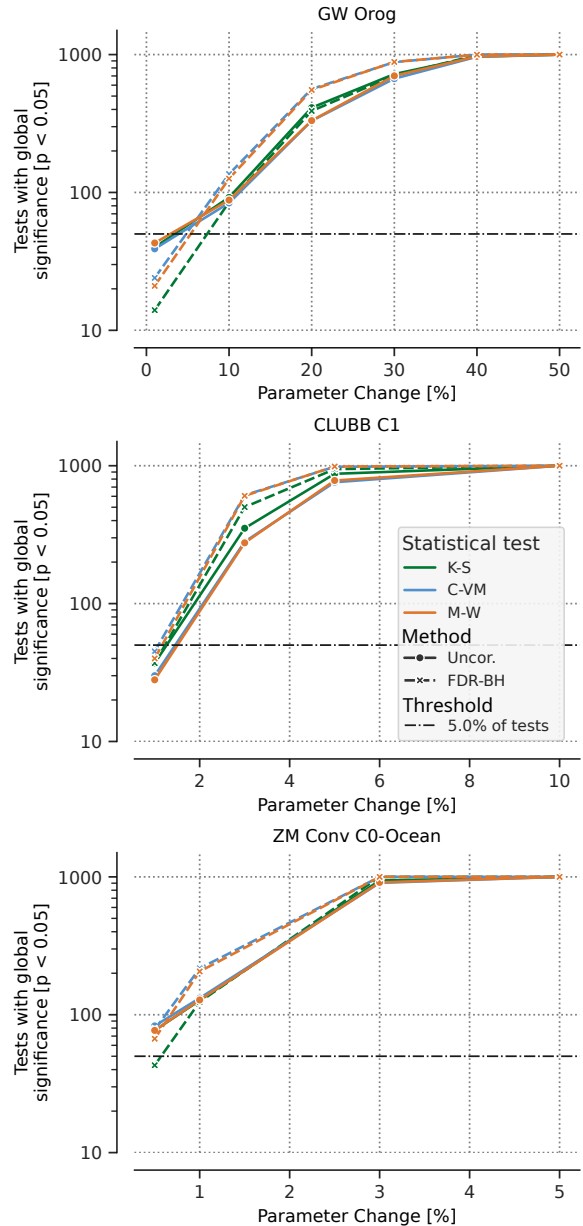

**Figure 4.** Number of bootstrap iterations that reject the global null hypothesis ($H_0$) out of 1000 bootstrap iterations for changes in tuning parameter `effgw_oro` (top), `clubb_c1` (middle), and `zmconv_c0_ocn` (bottom). Dashed black lines represent 5%–95% of all bootstraps.

and `zmconv_c0_ocn` parameters, there is an increase in power for all tests at all parameter changes, apart from the very smallest and largest percent changes (at the largest no power increase is possible as the uncorrected approach already rejects

all iterations). Both the C-VM and M-W tests have power increases for the `effwg_oro` parameter changes, but the K-S test has near 0 or power losses except at largest parameter changes.

For the smallest parameter change to `effgw_oro`, where MVK test exhibits greater power than the BH-FDR approach, it is possible that at the 1% change to `effgw_oro`, the simulated climate is not different in a meaningful way. This means that the spread of simulated climates (measured by global averages of the output fields) generated by adding a random perturbation to the temperature field does not differ significantly from the spread of the simulated climates with a 1% change to `effgw_oro`. Sampling errors may also be playing a role, given the small magnitude of change. Increasing the ensemble sizes can significantly increase the power of the tests as shown for MVK (Mahajan et al., 2019b), but operationally add to the computational cost. This trade-off between false negative rates and ensemble sizes is a decision left to model developers and code integrators. The power analysis here provides them with some reference to interpret the test results. For instance, if a non-bit-for-bit change passes the test (with the ensemble size of say, 30), developers can infer that its impact is likely smaller than a 5% change in C1 parameter of CLUBB, which can be detected at a high confidence by the tests. This contextual comparison helps determine whether to accept or investigate a change further and guides the selection of ensemble size needed to detect changes of interest. In the future, we will expand our power analysis to include other tuning parameter changes to better inform developers, integrators and domain scientists using the tests.

## 4.4 Optimization changes

Two simulation ensembles ("opt-O1" and "fastmath") were performed to apply the uncorrected and BH-FDR approaches to evaluate sensitivity of model results to compiler optimization flags, which are involved in the optimization of the underlying mathematics rather than model tuning parameters designed to account for varying physical processes. Bootstrapping procedures, similar to those described in the last section indicate that these optimizations do not have a significant impact on solution reproducibility (Figure 6). Between 10 - 31 out of the 1000 bootstrap iterations reject $H_0$ using the uncorrected statistical tests for both "opt-O1" and "fastmath" at the significance level of 0.05. After application of the BH-FDR approach, even fewer bootstrap iterations reject $H_0$, exhibiting a 0.5–0.7% chance to detect a change to "O1" and a 2.3–2.9% chance to detect a change in the "fastmath" flag. Reducing optimizations has been shown to yield climate reproducibility in previous studies as well (Baker et al., 2015; Mahajan et al., 2017), similar to our result that the simulated climate of "opt-O1" is statistically indistinguishable from that of the control ensemble that uses "-O3" optimizations. The aggressive optimizations enabled by "-fp-model=fast" are expected to decrease solution accuracy (Mielikainen et al., 2016; Büttner et al., 2024), however, in this set of ensembles they do not significantly alter the simulated climate. As previously mentioned, the "-fp-model=fast" option is already a default for several source files, thus this simulation ensemble tests its use only for those additional source files in EAM. This is different from testing the impact of using it across the model as a whole as in Mahajan et al. (2017). Acceptance of $H_0$ by the tests indicate that including the option for all of EAM, instead of selectively as is the default, does not result in statistically distinguishable solutions. This indicates that the "-fp-model=fast" optimization could be applied more generally throughout EAM code base. However, for higher resolutions or fully-coupled ensembles, this result may not apply as these

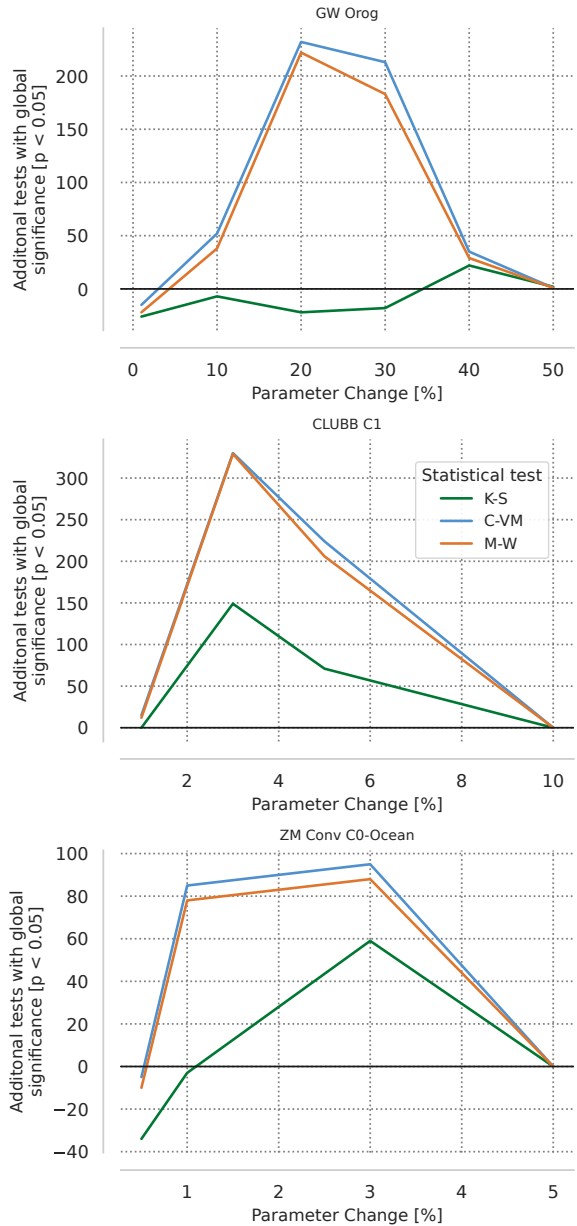

**Figure 5.** Additional number of bootstrap iterations that reject the global null hypothesis ($H_0$) out of 1000 bootstrap iterations for changes in tuning parameter `effgw_oro` (top), `clubb_c1` (middle), and `zmconv_c0_ocn` (bottom) when BH-FDR is applied for each statistical test, colors as in Fig. 4

configurations may respond to perturbations differently than ultra-low resolution model used here. Thus, further examination of its applicability may be required.

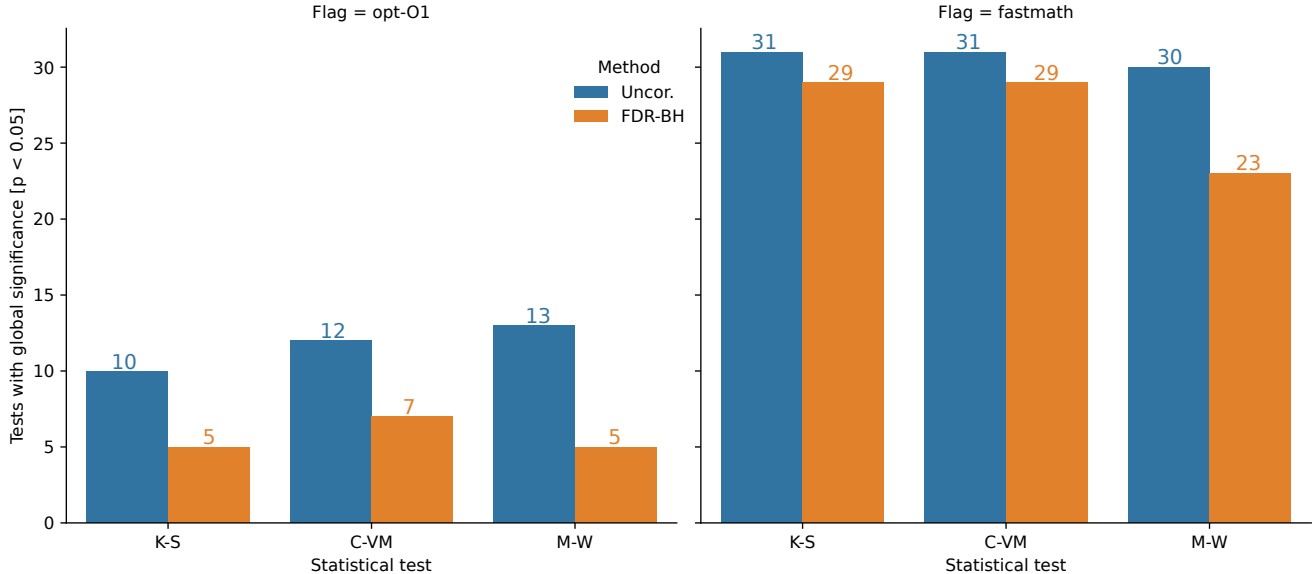

**Figure 6.** Number of bootstrap iterations that reject the global null hypothesis ($H_0$) for opt-O1 (left), and fastmath (right) ensembles, when compared to the control ensemble, out of 1000 bootstrap iterations for each statistical test, uncorrected in blue and B-H FDR (orange).

### 4.5 Standard resolution test

A test was also conducted using the USAF HPC11 using the same model version (E3SM v2.1), but at a higher resolution, called "standard-resolution" (called ne30pg2, $\approx 1.0°$ atmosphere) Two ensembles of 30 members each were constructed, a control ensemble with no parameter modification from default, and an ensemble where the `clubb_c1` parameter had been increased by 5% from 2.4 to 2.52. Table 3 details the number of output fields which are rejected at $\alpha = 0.05$ level for both uncorrected and BH-FDR corrected methods for each statistical test. Though a large ensemble was not conducted to determine an empirical global rejection threshold for the uncorrected tests, it is reasonable to assume that following the BH-FDR correction, that any field rejection results in a global null hypothesis rejection as in the "ultra-low" resolution ensembles. This means that for all tests, the two simulated climates are determined to be significantly different. Figure 7 shows that the map of grid box averaged cloud liquid amount is visually different on the order of $\pm 1 \times 10^{-6}\, kg\, kg^{-1}$, only one order of magnitude smaller than the field itself in the ensemble mean depending on the location. The largest differences appear mainly over the northern hemisphere, though smaller differences are noticeable in the Southern Ocean. Grid box-wise statistical tests are not conducted for this framework, but the field appears visually different in the mean sense, and is statistically distinct as confirmed by the statistical tests used.

The annual global means for each ensemble member are plotted against one another in figure 8. This visualizes differences in the distributions of the cloud liquid amount field which are used by the statistical tests to generate p-values. For a field which has no statistically significant differences in distributions between experiment (`clubb_c1 + 5.0%`) and Control, the quantiles

| Statistical test | Uncorrected rejections | BH-FDR corrected rejections |
|---|---|---|
| K-S | 29 | 22 |
| M-W | 32 | 27 |
| C-VM | 35 | 25 |

**Table 3.** Number of rejected fields in the "standard resolution" ensemble comparison for each statistical test, for uncorrected and BH-FDR corrected methods.

would fall along the 1:1 line in the Q-Q plot, as would the probabilities along the same line in the P-P plot, the histogram bars
would be of similar heights, and the cumulative distribution functions would have little distance between them. This is not the
case for the cloud liquid amount, and its local null hypothesis is rejected based on all tests and methods.

## 4.6 Ensemble size

The size of the sub-ensemble selected of 30 was chosen based on the previous results of Mahajan et al. (2022, 2019b) as a
balance between statistical power and computational efficiency. Figure 9 shows an increasing power for larger sub-ensemble
size selections, with the BH-FDR corrected approaches exhibiting larger power than the uncorrected counterparts. Here, the
choice of 30 ensemble member comparisons appears an appropriate balance between statistical power and computational time.
In the other figures of this study, the results are presented with an ensemble size of 30, to match the operational constraints so
results are directly applicable on the nightly testing.

## 4.7 Operational results

Fig. 10 shows the time series of $t$, or the number of variables rejecting $H_1$, for MVK and BH-FDR, for a period of a few
weeks after BH-FDR was implemented and included in nightly testing in late September last year. The additional statistical
tests were not performed on these ensembles, and are thus not able to be included here, as the nightly testing output is not
archived. The model maintained bit-for-bit reproducibility during these weeks. Bit-for-bit reproducibility is ascertained by a
suite of bit-for-bit tests that are run each night, testing the model under a variety of conditions. When these pass, the model
is bit-for-bit with previous results, and a test fail by MVK and BH-FDR on these days is thus known to be a false positive.
Fig. 10 shows that operationally BH-FDR has a reduced false positive rate of 1.9% (1 of 53 tests) as compared to 7.5% (4 of
53 tests) for MVK. $t$ varies between 0 to a maximum of 15 (on two individual days) for MVK, while $t$ is either 0 or 1 (on one
occasion) for BH-FDR. This solitary global null hypothesis rejection using BH-FDR does not occur at the same time as a global
rejection using uncorrected p-values, indicating that a systematic change in its statistics may not have occurred. In which case
both tests may be expected to fail. The BH-FDR approach yields a failing overall result when one particular p-value is very
small. In this case the `soa_a1_SRF`, a secondary organic aerosol field, was rejected very strongly, which meant the BH-FDR
method rejected the global null hypothesis. The rate of global rejections using MVK (7.5%) is higher than the targeted 5%,
which is reduced to under 2% using the FDR corrected p-value threshold, as is expected as the correction controls for the false

# Grid box averaged cloud liquid amount [kg/kg]

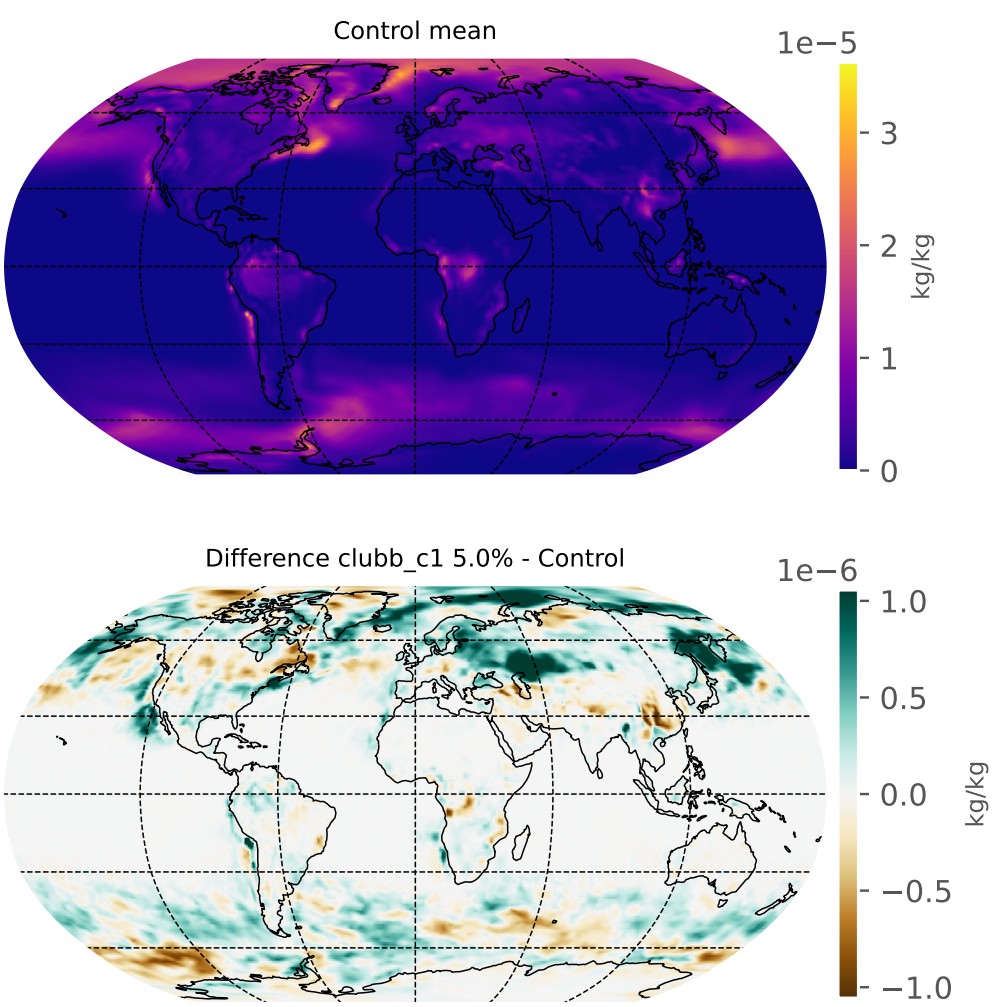

**Figure 7.** Map of ensemble mean cloud liquid amount for the ne30 resolution control ensemble (top) and the difference to the perturbed ensemble mean (bottom) in kg kg$^{-1}$.

discovery rate when using BH-FDR. The higher false positive rate of MVK can be associated with the small sample size (53 days of testing). In the future, we plan to expand the ensemble size of the control ensembles to derive the null distribution and evaluate its impact on operational false positive rates of MVK.

Operationally, the MVK test has recently been successful in identifying two bugs committed to the model's code, which were thought by developers to be non bit-for-bit, but not climate changing. The test returned a "fail" result following both changes,

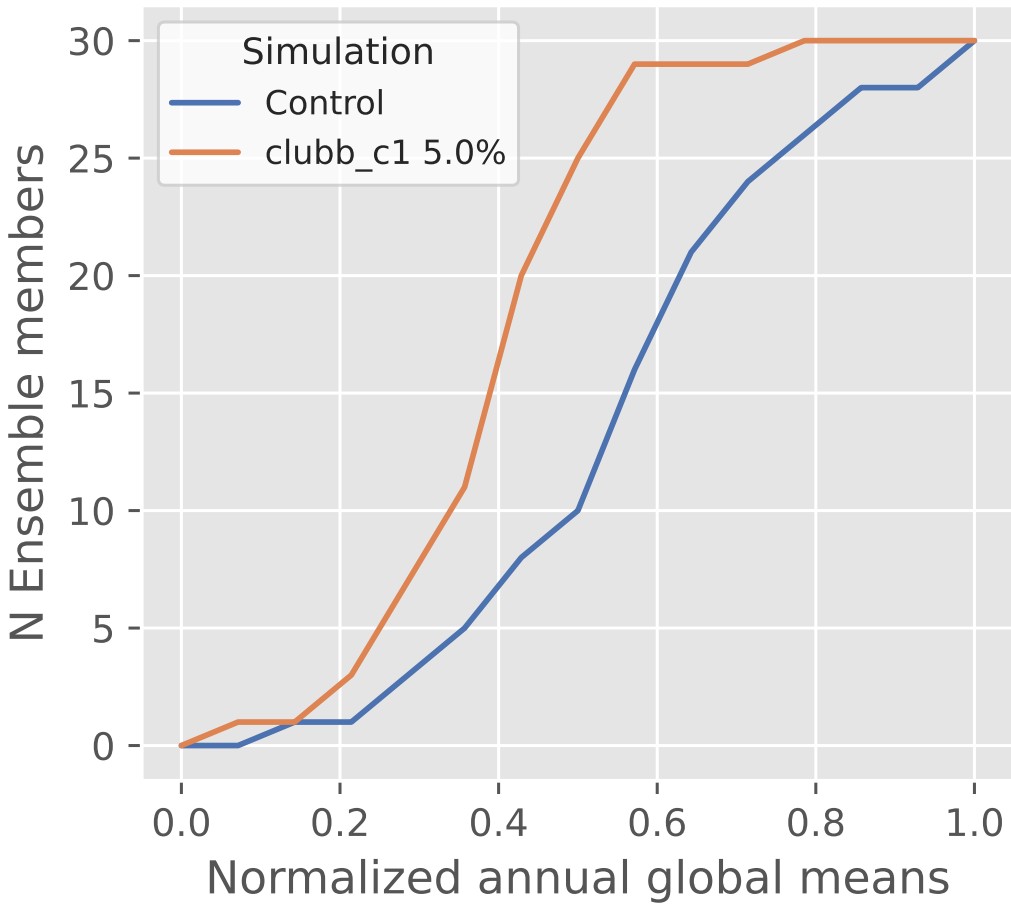

**Figure 8.** Cumulative distribution function for the cloud liquid amount field, as in Fig. 7. Both control and test ensemble annual global means are normalized together.

and the changes were reverted and eventually re-worked into non climate changing code alterations. This unintentional climate changing update was also captured by the BH-FDR approach and indicates its ability to capture erroneous alterations in addition to its ability to capture tuning parameter changes. In the first case, developers introduced changes related to aqueous-phase chemistry, specifically involving reactions tied to cloud water and trace gases. These updates were considered minor adjustments to internal model behavior and were not expected to alter the simulated climate. However, the tests flagged the update as a baseline failure, revealing statistically significant differences across many atmospheric variables. Further investigation showed that the change affected cloud-aerosol interactions and subsequently altered radiative balance, demonstrating how a seemingly minor, but non-bit-for-bit change in chemical processes led to broader simulated climate impacts. In the second case, ozone chemistry configuration changes were merged, aiming to improve numerical consistency in offline and

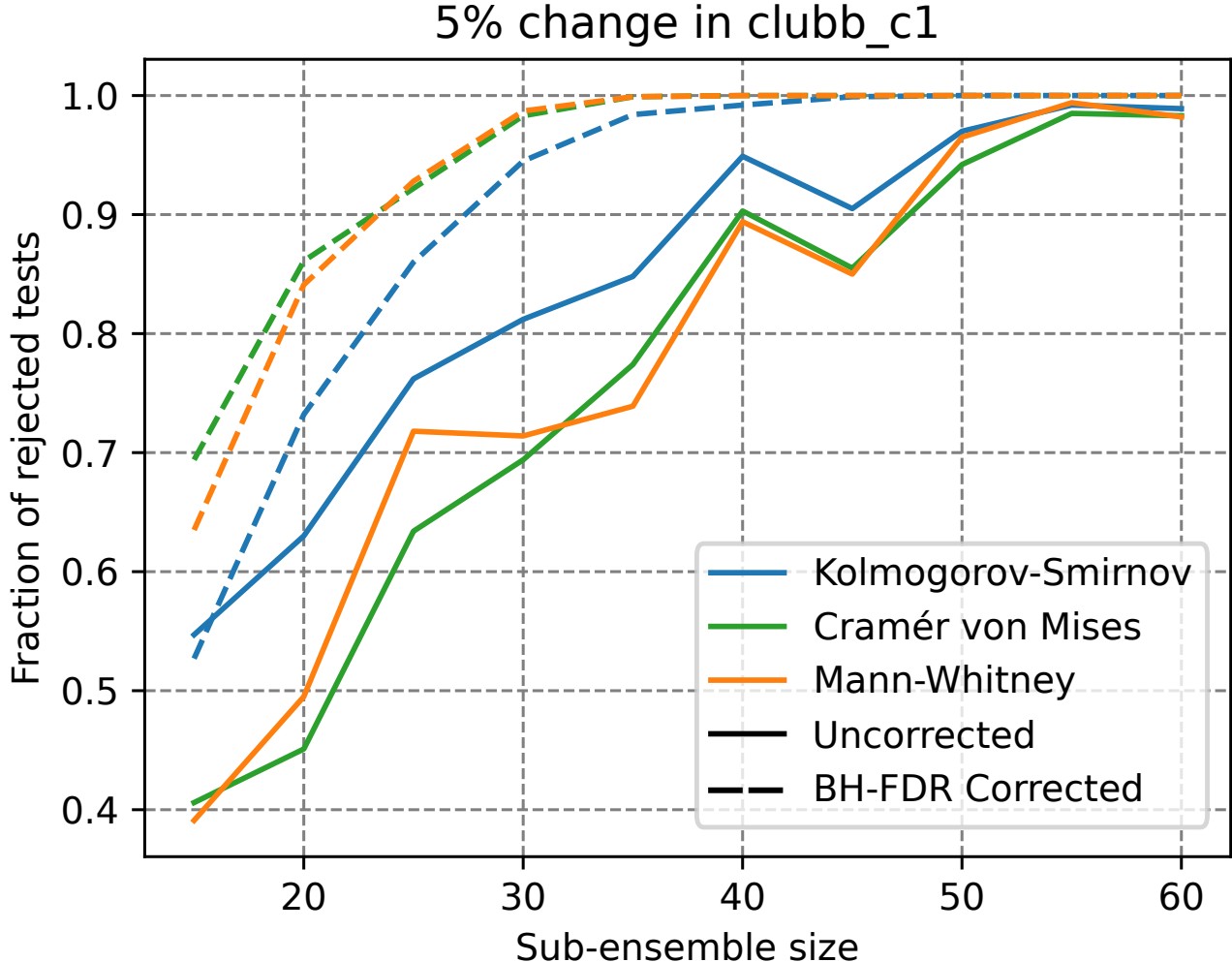

**Figure 9.** Power of statistical tests as in Fig. 4 on the `clubb_c1` 5% change to control ensemble comparison, but for varying sub-ensemble size selection (x-axis). Solid curves indicate uncorrected methods, dashed lines of the same color indicate BH-FDR corrected approach for each test.

online chemistry calculations and were also expected to be non-climate changing. However, MVK and BH-FDR detected a statistically significant difference, which was traced to elevated tropospheric ozone concentrations resulting from the changes.
The alteration impacted radiative forcing enough to shift the ensemble mean state, and was also subsequently corrected. These examples illustrate that the BH-FDR method has statistical power in the event of erroneous code changes in addition to its power in detecting perturbed parameters as does the MVK, thus its usefulness is not degraded.

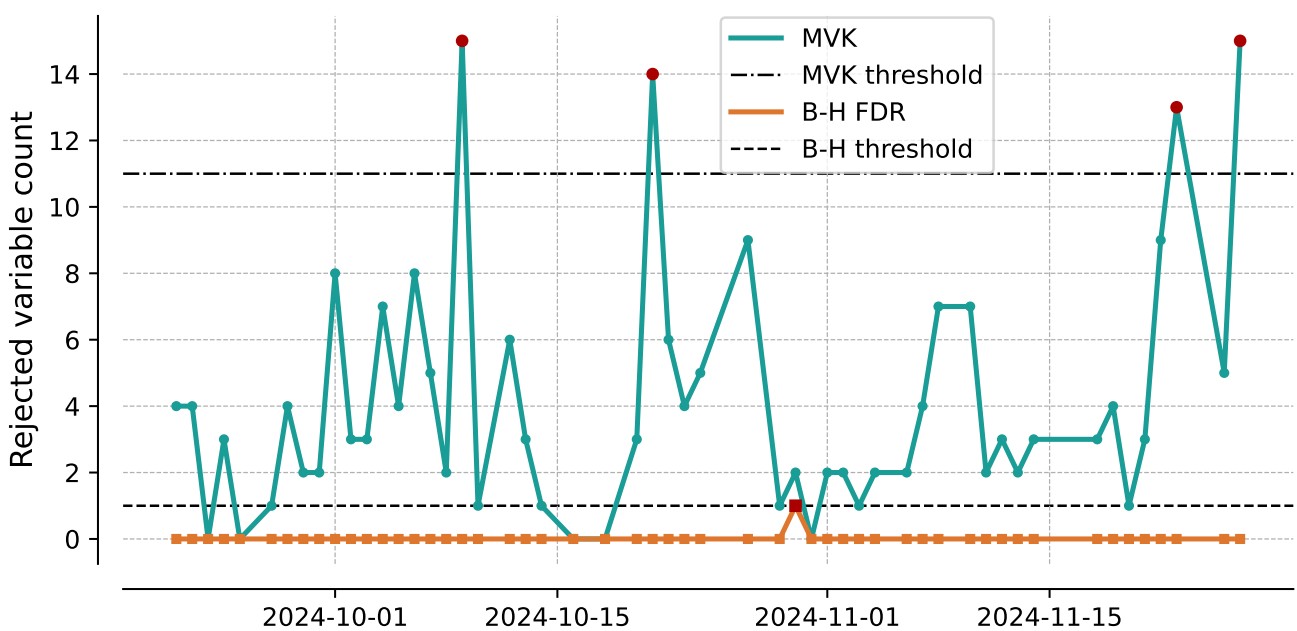

**Figure 10.** Number of rejected variables (of 120) for nightly tests of E3SM. The teal line shows rejection based on MVK p-values, orange shows number of rejections based on B-H FDR p-value thresholds.

## 5 Summary and Discussion

This study presents a new approach, BH-FDR, to evaluate statistical solution reproducibility of EAM after unintended non-bit-for-bit changes are introduced. BH-FDR improves on the existing MVK test and applies a false discovery rate correction to control Type I error inflation in multi-testing scenarios. While the original MVK approach relies on computationally expensive bootstrapping to determine critical value thresholds, BH-FDR offers a theoretically grounded and operationally simpler alternative. This computationally expensive threshold finding, where the ensembles represent ≈3700 node hours of computation time for ensemble generation, is demonstrated to be effectively eliminated, as the BH-FDR approach for all statistical tests can use a threshold of 1, while maintaining nearly the same, or improving statistical power.

Our evaluation using a comprehensive suite of ensembles, including both parameter perturbations and compiler optimization changes, demonstrates that BH-FDR approach maintains or improves the statistical power of the MVK test, while significantly reducing false positive rates. Notably, the BH-FDR approach eliminates the need for re-calibrating critical value thresholds after major model revisions. Operational implementation of this method in nightly E3SM testing has further validated its utility, showing a reduction in false positives from 7.5% to 1.9%. Overall, the BH-FDR approach enhances the robustness, accuracy, and efficiency of statistical testing for climate model reproducibility.

Additional tests were performed with the Cramér-von Mises and Mann-Whitney U tests to add a broader perspective on the applicability of the BH-FDR method. For uncorrected methods, the K-S test had the highest power and equivalent false positive rates to the C-VM and M-W tests, but in using the BH-FDR method, the power of those tests increased beyond that of the K-S test. Primarily, this indicates that the BH-FDR method works well for multiple different statistical tests and is valid so long as the underlying statistical test is valid for the data being examined. We plan to add these additional statistical tests to the nightly suite and continue to examine ways to usefully combine the results of these different tests to evaluate if two simulation ensembles are statistically similar.

We also examined the sensitivity of the results to ensemble size selection. Part of the procedure applied here to find the power of the various tests involves comparing two sub-ensembles each drawn from two larger ensembles at random. The power of all three statistical tests both for uncorrected and BH-FDR corrected methods scales with sub-ensemble size. There is, however, a trade-off between computational efficiency and statistical power. As an example, in this configuration with "ultra-low" resolution on the Chrysalis machine, one ensemble member runs on a single node in approximately 15 wall-seconds per model day or about 1 hour 45 minutes per 14-month simulation. Each additional ensemble member then adds a node to the computational requirements, so a 30 member ensemble uses 30 nodes for 1 hour 45 minutes, or approximately 53 node-hours. We believe that an ensemble size of 30 strikes the right balance between statistical power and computational costs for routine nightly testing, but a developer is able to choose to increase this size to enhance the statistical power as needed.

A caveat of our study here is that BH-FDR (and MVK as well) has been rigorously applied to and evaluated for the ultra-low resolution version of the model, which is not used in practical applications, and only a single test was performed at a higher resolution. The consistency of test results across two different model resolutions provides evidence that the test results with the ultra-low resolution model hold for standard and high resolution model configurations used for production runs. Higher resolution models resolve finer scale processes, which can effect numerical sensitivity, internal variability and process feedbacks of the model. In the future, we plan to identify the underlying reasons and scenarios in which test results may or may not remain consistent across resolutions. Nonetheless, an earlier unpublished work found that the results of MVK applied to ultra-low resolution ensembles and MVK applied to standard resolution ensembles were consistent when evaluating a port of an earlier version of E3SM to a new machine at the National Energy Research Scientific Computing Center (NERSC). Further, we will explore enhancing computational feasibility of the tests by using shorter run times of simulation ensembles (Milroy et al., 2018), allowing for routine testing with higher resolution models.

In addition to its application in traditional Earth system models, the BH-FDR-based statistical testing framework has potential for assessing the reproducibility of AI-based climate models, which are rapidly gaining prominence in climate science. Recent developments such as ClimateBench (Watson-Parris et al., 2022), FourCastNet (Pathak et al., 2022), and Pangu-Weather (Bi et al., 2022) demonstrate the capabilities of deep learning models to emulate or replace components of physics-based models with significant computational advantages allowing the generation of very large ensembles at very low computational costs. However, verifying the reliability and reproducibility of these models presents unique challenges. AI models often involve stochastic elements in training, sensitivity to floating-point precision, and reliance on hardware-specific optimizations, all of which can lead to variability in output across runs. Standard bit-for-bit reproducibility tests are inadequate in this con-

text, and statistical frameworks like MVK or BH-FDR could serve as robust alternatives to assess whether differences in AI model outputs are statistically meaningful or within expected variability. Prior work has highlighted the need for principled evaluation methods tailored to the probabilistic nature of machine learning in scientific applications (Rasp et al., 2020; Dueben et al., 2021), and integrating ensemble-based hypothesis testing into AI model workflows could be a step toward more rigorous, interpretable, and trustworthy deployment of AI systems in operational climate modeling. Ultra-low resolution models, like the one used here, can also be used to create very large ensembles at low computational cost allowing comparisons with their AI surrogate large ensembles. We plan to conduct such evaluations in the near future.

*Code availability.* The code for this work can be found at https://github.com/mkstratos/detectable_climate (Kelleher and Mahajan, 2025a)

*Data availability.* The bootstrap data from each comparison is available in (Kelleher and Mahajan, 2025b)

*Author contributions.* MK and SM developed the methodology. MK wrote the code, conducted the simulations, and wrote the first draft of the manuscript. SM supervised the project and contributed writing to the final manuscript.

*Competing interests.* The contact author has declared that none of the authors has any competing interests.

*Disclaimer.* Any subjective views or opinions that might be expressed in the paper do not necessarily represent the views of the U.S. Department of Energy or the United States Government.

*Acknowledgements.* This research was supported as part of the Energy Exascale Earth System Model (E3SM) project, funded by the U.S. Department of Energy, Office of Science, Office of Biological and Environmental Research. The authors also gratefully acknowledge the computing resources provided on Blues, a high-performance computing cluster operated by the Laboratory Computing Resource Center at Argonne National Laboratory. This research used resources of the Oak Ridge Leadership Computing Facility, which is a DOE Office of Science User Facility supported under Contract DE-AC05-00OR22725. The authors also acknowledge the numerous open-source libraries on which this work depends, Harris et al. (2020); Virtanen et al. (2020); Hoyer and Hamman (2017); Dask Development Team (2016); Wes McKinney (2010); Hunter (2007); Waskom (2021); Seabold and Perktold (2010). The authors also acknowledge the seven anonymous reviewers, their comments have made this a more robust investigation.

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
