# Peer review of "Enhanced Climate Reproducibility Testing with False Discovery Rate Correction"

_EGUsphere, 2025_

## Author Comment (AC1)

**1 Introduction**

The authors thank the numerous reviewers for their time in reviewing and constructing helpful comments, which will lead to a greatly improved manuscript. In general, the comments focused on the wider applicability of the testing method, and the robustness of the results, each of which will be addressed in the point by point responses. In these responses, we will focus on the major or specific comments, as minor comments, language fluidity, and typos will be addressed after revisions.

**2 Responses**

**2.1 Reviewer 1**

- *"I think it would be worth adding further explanation of the correction in section 2.2, as it is a crux of understanding the work. I had to read through a few times and revisit this section to understand what is going on. I might also suggest modifying equation 2.2, as it currently reads as the maximum of a set of booleans. Would argmax be a better fit? It would be helpful, I think, to give a plot of p(i) values and the calculated PFDR value for the control ensemble at least."*
  **This is an excellent point as it is the basis on which the rest of the paper is based, we will further refine this section and the equation, as suggested here and by other reviewers. We will also add an explanation figure which shows how the p-values are corrected.**

- *"Does the estimate of p(i)'s and PFDR stay stable with different ensemble sizes? In other words, how does one know that the 30-member ensemble is sufficient? A similar question would apply to the estimates of the false positive rate and sensitivity. As a model changes, is there a point where a different ensemble size is necessary, and how would one know?"*
  **We expect the values of $p_{FDR}$ to change for different ensemble sizes, and for larger ensemble sizes to result in greater statistical power, up to a point (see figure 3 of Mahajan [2021]). We will perform further bootstrap analyses, varying ensemble size to show evidence for this hypothesis.**

- *I think the information in Figure 1 is very interesting. It could be worth clarifying further. I read that description as meaning that for each modification, a base ensemble and a test ensemble are created. Versus testing fastmath against the control?*
  **The results presented in figure 1 are ensemble self-comparisons, i.e. for each 120 member ensemble generated, which includes the control ensemble, the perturbed parameter ensembles, and the compiler optimization flag ensembles, it is bootstrapped against itself. Meaning two random 30 member draws from the same ensemble are compared against each other 1000 times, which we use to establish a threshold defining the expected number of fields which can be rejected while maintaining the same simulated climate.**

- *In table 2 I would include the PFDR values for each ensemble.*
  **This information will be added to the table.**

- *I found the use of the 95 percentile in Figure 2, and its description in the paragraph at line 215, to be confusing. For instance, with a 10% parameter change of GW Orog with MVK, does this mean it would be possible 94% of bootstrapped tests don't return a failure? Also, I believe the horizontal lines might be mislabeled in the caption and legend. Why not just plot the mean number of failed variables with bars for the std? Perhaps there is a good reason for*

*the percentile that I am missing, or perhaps there is some mismatch between the paragraph at 215 and what the plot is actually showing.*

**We will add further clarification to the text describing this, but the main point of this figure is to show that 95% of the 1000 bootstrap comparisons between the control and for example the 10% change to `effgw_oro`, have 12 of 117 output fields rejected for MVK p-values, and 1 of 117 output fields rejected for BH-FDR corrected p-values. The 95% level is chosen here as it is the inverse of our significance level $\alpha = 0.05$, and as such one could expect to see 5% of bootstrap iterations failing the test on random chance, thus if fewer than 5% have passed, this is a strong indicator that the two ensembles are significantly different. We use the percentiles here and throughout the manuscript as they are robust and resistant measures of the data when compared to mean / standard deviation [Wilks, 2019].**

- *I find it interesting and might be worth discussing, in regard to Figure 5, that there doesn't seem to be any correlation of between the false positives for MVK and BH-FDR approaches. On the one test day that fails BH-FDR the MVK approach is nowhere near failing.*
  **We believe this to be caused by the MVK approach not taking into account the value of p-values, only that there are enough below the threshold. The BH-FDR approach can yield a failing overall result if one particular p-value is very small.**

**2.2 Reviewer 2**

- *While the adoption of the MVK test is motivated by previous studies cited in this manuscript, there are alternatives to the MVK test that could also be considered before applying an FDR correction. For instance, the Anderson-Darling or Cramér-von Mises tests have advantages over the Kolmogorov-Smirnov test, such as being more sensitive to repeated deviations from the empirical distribution functions, detecting deviations in the tails of the distributions, and effectively handling skewed distributions.*
  **The MVK / Kolmogorov-Smirnov test was used here as it was used in the production nightly testing environment, but this is an excellent point. Further analysis will be performed using additional initial statistical tests, then performing the correction on each set of p-values.**

- *As I understand it, the MVK test is performed on two independent ensembles with N=30 members, comparing the distribution of annual global means for 120 output fields (local null hypothesis). What is the rationale for choosing N=30 specifically, rather than other values of N? The reported results may vary with different values of N. In addition to the caveat of using a "low ultra-resolution model," the reported statistical results are conditional on a very specific value of N. It would be important to explore lower and higher N values to provide evidence of the effect or perhaps non-effect of the FDR correction under different sample size scenarios.*
  **The $N = 30$ choice was based on earlier results from Mahajan [2021] where an ensemble of N=30 was found to balance statistical power and computational time. We will explore this by varying this parameter during the boostrapping phase, which is expected demonstrate that a larger ensemble size will exhibit greater power, but that it is offset by the computational cost.**

- *Furthermore, why are "annual global means" the only statistic of interest rather than regional means or perhaps extreme value statistics, such as the maximum or minimum values over the same time-period? Testing for equality of distributions of global means represents just one focused statistical aspect of a set of climate ensembles with hundreds of output fields. I believe that many other tests beyond those proposed in this paper could be considered and of interest, potentially leading to different statistical assessments of the ensembles and a wide range of*

*results. Even under a coarse spatial resolution, it is to be expected that tests of a local mean or some other local statistic may lead to a different assessment than those resulting from a global statistic. This raises the question of what does it mean to study and test "climate reproducibility"? That variations of the model simulations provide similar global means?*

**We appreciate this comment and agree that regional means, local diagnostics, and extremes can provide valuable perspectives on ensemble behavior. In this work, however, we focus on annual global means, following previous work [Baker et al., 2015, Mahajan et al., 2017]. Atmospheric variability is highly heterogeneous in space and time, with sharp gradients and short-term fluctuations that complicate regional analyses. Global averaging suppresses this weather-driven noise and reduces dimensionality, making it more suitable for applying the ensemble consistency framework. For the ocean model component, where the ocean is more homogeneous we have implemented a test that accounts for spatial variability as well conducting tests at each grid point [Mahajan, 2021]. Baker et al. [2016] also use a similar approach and note that while global averaging can obscure signals in the ocean, it remains effective and appropriate for atmospheric applications. Further, extremes require much larger ensembles to be reliably assessed. Mahajan et al. [2017] demonstrated that with ensembles of about sixty members, extremes of temperature and precipitation were often statistically indistinguishable even when mean distributions diverged, underscoring their limited sensitivity in this context.**

- *The false positive rates and power analysis presented in the manuscript consider a one-at-a-time variation of the tuning parameters. Should simultaneous variations be considered to study interactions between the "effgw_oro" and "clubb-c1" parameters and understand how these interactions impact the rates and power? I realize that considering all possible interactions may create a large computational burden. However, selecting a few potential interactions of interest could provide further evidence in the application of an FDR correction to assess the reproducibility of climate simulations.*

  **We have considered just two tuning parameters here out of hundreds of available adjustments. Varying the two together may indeed yield interesting results, and could be indicative of how the test responds to a variety of changes, but we feel this is out of the scope of this work to investigate, and chose to vary them individually for the sake of simplicity. Additionally, using the global mean as the test statistic of interest makes the output of this test easier to understand at a glance by model developers in an operational setting and if further understanding is needed then additional tests can be performed.**

**2.3   Reviewer 3**

- *Limited Generalizability: The study's findings are highly specific to:*

  - *A single model (E3SM v2.1) at ultra-low resolution ( 7.5° atmosphere)*
  - *Annual global means of standard output variables*
  - *A particular computational environment (Argonne's Chrysalis machine with Intel compiler)*

  *The authors acknowledge some limitations but do not adequately demonstrate that their approach generalizes beyond this narrow context. Testing at production resolutions, with other models, or with different climate statistics (regional patterns, extremes, seasonal cycles) is essential to support claims of broader applicability.*

  **We acknowledge the limited scope of this work, as an iterative development on**

existing operationally used tools. The applicability of these results to production resolutions will be addressed by comparing two 30 member ensembles at the NE30 ($1°$) horizontal resolution. Testing this method in other Earth system models is beyond the scope of this manuscript and as mentioned above, global means are used to simplify and make the results accessible to model developers.

- *Statistical Assumptions and Dependencies: The paper inadequately addresses:*

  - *The correlation structure among the 117 tested variables, which violates the independence assumption of standard BH-FDR*
  - *Whether the "theoretical" critical value of 1 for rejecting the global null hypothesis holds beyond their specific ensemble configuration*
  - *The choice of $q^* = \alpha = 0.05$ for FDR control without justification*

It is an excellent point that the statistical dependence of the 117 output fields may violate the assumptions of the BH-FDR correction, we will explore this possibility in the updated manuscript. The BH-FDR correction is applied to the p-values resulting from the comparison of the global annual mean of each field from two 30 member ensembles, thus the statistical independence of these random variables is what will be tested, i.e. the correlations of the annual mean each output field against each other output field.
We will add justification for using $q^* = \alpha = 0.05$, though this is the common practice to assign the false discovery rate to be the same as $\alpha$ [Wilks, 2016, Ventura et al., 2004].

- *Incomplete Methodological Comparisons: The manuscript lacks crucial comparisons with alternative approaches:*

  - *Permutation testing: While the authors mention that MVK "was found to give similar results as compared to permutation testing" (citing Mahajan et al., 2019b), they do not actually compare BH-FDR against permutation-based methods for their specific application.*
  - *Alternative multiple testing corrections: No comparison with other FDR methods (e.g., Benjamini-Yekutieli for dependent tests) or other approaches like the False Discovery Exceedance method.*

Additional statistical tests will be performed, as mentioned in our responses to other reviewers, and we expect that the results of a permutation based approach will track well with the permutation based testing of the MVK (uncorrected) method as the corrections operate on the p-values after the bootstrapping (or permutation) takes place.
Text will be added to describe the previous examination of other correction methods, including Benjamini-Yekutieli and Bonferroni methods, which yielded lower power than both the MVK and BH-FDR approaches for this set of experiments. These results match with those of Ventura et al. [2004], finding that the assumptions in Benjamini and Yekutieli [2001] were too restrictive, while Benjamini and Hochberg [1995] was simple to compute and held up for dependent data.

- *Limited Scope of Reproducibility Testing: Testing only annual global means represents a minimal assessment of climate reproducibility. Climate models are evaluated on their ability to simulate regional patterns, variability, extremes, and trends. A model could pass this test while having significant regional biases or incorrect variability patterns.*
The global means as a test of climate reproducibility is intended a first-order test, as mentioned above in response to Anonymous Reviewer 2.

- **Minor Comments**

    - *The power analysis covers only two parameters. Testing additional parameters with different sensitivities would strengthen the conclusions.*
    An additional tuning parameter from a third physical parameterization will be added to the two parameters examined.

    - *The operational validation period (53 days) is quite short for drawing robust conclusions about false positive rates.*
    This is true, the operational period was chosen as it was a period where the model code base passed bit-for-bit testing for this configuration, and could not be expanded. It is intended as a real-world operational example where code changes were applied rather than a controlled experiment where tuning parameters were varied.

    - *The manuscript reads more as a technical note about an incremental improvement to E3SM's testing infrastructure rather than a methodological advance in climate model evaluation*
    This point is well taken, we hope additional tuning parameters, simulation resolutions, and statistical tests will highlight the broader applicability of the test as it is used.

**2.4  Reviewer 4**

- *I believe the paper effectively demonstrated that the false negative rate is lower while the false positive rate is nearly unchanged for BH-FDR compared to MVK in this specific use case. My biggest issue with the manuscript is that it seems to largely be a method development paper for ensuring statistical stability of a numerical model with an Earth system model used as the exemplar. Even though this is for a special issue focused on ensemble design, I think the article need more of an explicit relation to Earth System Dynamics. So, while I believe the paper has strong merits for publication, I am not sure if this is the correct journal. Perhaps a more statistics-oriented journal would be appropriate.*
**As highlighted in our response to Anonymous Reviewer 3, we feel the broader applicability of this manuscript will be highlighted by the inclusion of different resolutions, tuning parameters, and statistical tests.**

- *It's not clear to me what Section 4.4 is adding to the paper. They aren't showing any meaningful difference in the climate produced by two alternative compilers, but as they rightfully point out this is likely influenced by the fact that they are running ultra-low resolution simulations. So, they can't really comment on the ultimate effect of these compilers on the mean state of the climate outside of this specific configuration, which feels like a weak result.*
**The results in this section are from using different compiler optimization flags to the same (Intel) compiler. It is intended as a way to introduce potential differences to the ensembles without the use of tuning parameters as in Baker et al. [2015], Mahajan et al. [2017]. In the updated manuscript, we will more carefully describe the changes and their expected results.**

- *Minor comments*
The text corrections will be applied to the updated manuscript.

**2.5  Reviewer 5**

- *There were repeated comments regarding the potential limitations of a 120-member ensemble in this work suggesting that this may not be enough simulations to fully represent the internal variability of MVK. While I found the rest of the paper to be a compelling defense of the update, this comment stuck with me and introduced a level of uncertainty into the conclusions that I*

*was uncomfortable with. I would like to see some justification or supplementary analysis that suggests that this ensemble is sufficient for these results or at least can give some estimated range of uncertainty.*

**This is an excellent point, and it will be addressed in the updated manuscript. An Earth system model's internal variability is difficult to diagnose, and to determine where saturation is. We plan to add additional ensemble members to the control ensemble to further evaluate if the range of variability is captured.**

- *Can you include a statement in your paper or in the results section that describes the improvement in computational time with BH-FDR versus MVK? This would then defend the twofold benefits of the BH-FDR that were described in the introduction of your paper.*
  **We will more carefully describe this improvement, as the BH-FDR approach is not an improvement in computational time over the traditional MVK approach, as it is an additional step on top of the MVK approach. The improvement in computational time comes from not needing to perform the multi-ensemble bootstrapping required to set the empirical threshold used to reject the global null hypothesis.**

- *Minor comments*

  - Corrections and suggestions will be applied to the update manuscript
  - *Line 177: While this method intuitively seems robust, is there justification for the 1000 times sample?*
    The 1000 bootstrap iterations was somewhat arbitrary, but chosen as a large enough to have statistically significant results, based on earlier work in Mahajan et al. [2017], which used 500 iterations.
  - *On the topic of detectability and mainly a question of personal interest: based on the relative sensitivities, can you generalize the detectability of a parameter change based on its sensitivity?*
    **This is an interesting question, and we have not explored this possibility. Based on the two parameters examined here, they do seem closely related. An additional perturbed parameter ensemble, as mentioned in responses to other reviewers, may offer additional insight into this, though further examination would be left to future work.**

**2.6 Reviewer 6**

Here we will respond to specific comments.

- *the assumptions of the Benjamini–Hochberg False-Discovery-Rate should be explicitly stated and checked*
  **The general assumption of p-value independence will be examined, though the work of Benjamini and Yekutieli [2001] broadens the assumptions to correlated data, which was also tested in Ventura et al. [2004].**

- *the work should be put in a more general context, including*
  *https://gmd.copernicus.org/articles/15/3183/2022/*
  *and the previous work*
  *https://dl.acm.org/doi/pdf/10.1145/3468267.3470572*
  *The former work could, in particular, help to comment on some choices made (e.g. only Kolmogorov-Smirnov and the choice of ultra-low resolution) even though the tests are different.*
  **Adding the context of Zeman and Schär [2022] and Mahajan [2021], will enhance the applicability of this technique to additional testing frameworks, as our scope is currently limited to the E3SM Atmosphere Model.**

- *It seems that FDR is also used in one of the author's previous papers:*
  *https://dl.acm.org/doi/pdf/10.1145/3468267.3470572*
  *and it should be made clear how the current results differ from the one in the previous paper.*
  **Further context will be added to the updated manuscript, but briefly, the FDR correction in Mahajan [2021] was applied to the p-values computed at each grid point in each 2- or 3-D output field, rather than the p-values of global means of the output fields, and was designed specifically for the MPAS-Ocean model.**

- *The case study is quite restricted. Possibly more cases could be considered. I also agree with the other reviewers that sometimes more explanation could be given on the choice of methods/parameters.*
  **As mentioned in responses to other reviewers, further parameters, resolutions, and statistical tests will be examined to further the broader applicability of the manuscript.**

**2.7 Reviewer 7**

- *There is a lot of jargon in the paper in the introductory sections, which makes it harder to read for a non-statistician. Some of the definitions (like Type I errors) were not clear from the text (even though the text refers to Sec. 4 for Type I error definition, there is no further clarification in Sec. 4). I found some clarifications in Mahajan2017 and Mahajan2019, but I think it is ok to repeat definitions of basic concepts in this paper.*
  **In the updated manuscript we will take care to detail and reduce the level of statistical jargon, especially in the introductory sections.**

- *Equation 1 – maximum function has boolean input, instead of a number. Is there a typo? Also, why does p-level there depend in an index ("i") of a field to be tested? The authors list m=117 above, but in general m can be arbitrary large.*
  **This equation will be clarified in the updated manuscript, it should indicate that $p_{FDR}$ is the maximum p-value for which the inequality is satisfied. The p-values are indexed $1...m$, indicating the field from which it is computed, then sorted from smallest ($i = 1$) to largest ($i = m$).**

- *It would be very interesting to see (if not in this publication, then maybe in future ones) which simulations in the new framework (and possibly the old one) are deemed climate-changing by looking at their plots (standard latlon mean fields, zonal means, etc.), and evaluate them by an eyeball norm. Would a human expert also consider these simulations "climate changing" or not? Something similar is presented in Mahajan2017, but not exactly. There is an issue that the current work was done with an ultralow res E3SM (ne4 for the atmosphere), and this configuration of E3SM might not have established climatologies.*
  **This is certainly possible, though the detectable differences here are statistically significant, it may not be apparent on these maps, as the differences are relatively small in the absolute sense. We will explore this, and the updated manuscript will include an example figure if it is illustrative.**

**References**

A. Baker, D. Hammerling, M. Levy, H. Xu, J. Dennis, B. Eaton, J. Edwards, C. Hannay, S. Mickelson, R. Neale, et al. A new ensemble-based consistency test for the community earth system model (pyCECT v1. 0). *Geoscientific Model Development*, 8(9):2829–2840, 2015. doi: 10.5194/gmd-8-2829-2015.

A. H. Baker, Y. Hu, D. M. Hammerling, Y.-H. Tseng, H. Xu, X. Huang, F. O. Bryan, and G. Yang. Evaluating statistical consistency in the ocean model component of the community earth system model (pyCECT v2.0). *Geoscientific Model Development*, 9(7):2391–2406, 2016. doi: 10.5194/gmd-9-2391-2016.

Y. Benjamini and Y. Hochberg. Controlling the false discovery rate: A practical and powerful approach to multiple testing. *Journal of the Royal Statistical Society: Series B (Methodological)*, 57(1):289–300, 1995. doi: 10.1111/j.2517-6161.1995.tb02031.x.

Y. Benjamini and D. Yekutieli. The control of the false discovery rate in multiple testing under dependency. *The Annals of Statistics*, 29(4):1165–1188, 2001. ISSN 00905364, 21688966. URL http://www.jstor.org/stable/2674075.

S. Mahajan. Ensuring statistical reproducibility of ocean model simulations in the age of hybrid computing. In *Proceedings of the Platform for Advanced Scientific Computing Conference*, PASC '21, New York, NY, USA, 2021. Association for Computing Machinery. ISBN 9781450385633. doi: 10.1145/3468267.3470572.

S. Mahajan, A. L. Gaddis, K. J. Evans, and M. R. Norman. Exploring an ensemble-based approach to atmospheric climate modeling and testing at scale. *Procedia Computer Science*, 108:735–744, 2017. ISSN 1877-0509. doi: 10.1016/j.procs.2017.05.259. International Conference on Computational Science, ICCS 2017, 12-14 June 2017, Zurich, Switzerland.

V. Ventura, C. J. Paciorek, and J. S. Risbey. Controlling the proportion of falsely rejected hypotheses when conducting multiple tests with climatological data. *Journal of Climate*, 17(22):4343 – 4356, 2004. doi: 10.1175/3199.1.

D. S. Wilks. "the stippling shows statistically significant grid points": How research results are routinely overstated and overinterpreted, and what to do about it. *Bulletin of the American Meteorological Society*, 97(12):2263 – 2273, 2016. doi: 10.1175/BAMS-D-15-00267.1.

D. S. Wilks. Chapter 3 - empirical distributions and exploratory data analysis. In D. S. Wilks, editor, *Statistical Methods in the Atmospheric Sciences (Fourth Edition)*, pages 23–75. Elsevier, fourth edition edition, 2019. ISBN 978-0-12-815823-4. doi: 10.1016/B978-0-12-815823-4.00003-1.

C. Zeman and C. Schär. An ensemble-based statistical methodology to detect differences in weather and climate model executables. *Geoscientific Model Development*, 15(8):3183–3203, 2022. doi: 10.5194/gmd-15-3183-2022.

---

## Author Response (AR1)

**1   Introduction**

The authors thank the numerous reviewers for their time in reviewing and constructing helpful comments, which we feel has lead to a more robust and broader study and improved manuscript. These responses focus on the major or specific comments, while the minor textual changes have been applied, and are updated from the previous responses to reflect the updated manuscript.

**2   Responses**

**2.1   Reviewer 1**

- *"I think it would be worth adding further explanation of the correction in section 2.2, as it is a crux of understanding the work. I had to read through a few times and revisit this section to understand what is going on. I might also suggest modifying equation 2.2, as it currently reads as the maximum of a set of booleans. Would argmax be a better fit? It would be helpful, I think, to give a plot of p(i) values and the calculated PFDR value for the control ensemble at least."*
**This has been re-framed to the mathematical equivalent perspective of adjusting the critical $\alpha$ threshold rather than p-values, and a new plot has been added (Eq. 1, Sec. 2.4 and Fig. 1).**

- *"Does the estimate of p(i)'s and PFDR stay stable with different ensemble sizes? In other words, how does one know that the 30-member ensemble is sufficient? A similar question would apply to the estimates of the false positive rate and sensitivity. As a model changes, is there a point where a different ensemble size is necessary, and how would one know?"*
**The $\alpha_{FDR}$ does change as the number of p-values which are rejected change with an increase in ensemble size. This is now discussed in Sec. 4.6, and Fig. 9. The larger ensemble sizes do indeed have more power, but as discussed in the manuscript there is a trade-off in computational cost.**

- *I think the information in Figure 1 is very interesting. It could be worth clarifying further. I read that description as meaning that for each modification, a base ensemble and a test ensemble are created. Versus testing fastmath against the control?*
**We have updated the description of the data which goes into this figure, (now figure 2) to more clearly describe this as a threshold-finding self comparison.**

- *In table 2 I would include the PFDR values for each ensemble.*
**Since $\alpha_{FDR}$ changes with each bootstrap comparison, it to put that into a table showing the bulk metrics of each ensemble, as a mean or median of this metric won't add much insight beyond that presented in figures 3 and 4 in our opinion.**

- *I found the use of the 95 percentile in Figure 2, and its description in the paragraph at line 215, to be confusing. For instance, with a 10% parameter change of GW Orog with MVK, does this mean it would be possible 94% of bootstrapped tests don't return a failure? Also, I believe the horizontal lines might be mislabeled in the caption and legend. Why not just plot the mean number of failed variables with bars for the std? Perhaps there is a good reason for the percentile that I am missing, or perhaps there is some mismatch between the paragraph at 215 and what the plot is actually showing.*
**Further clarification was added to the text in section 4.3, starting on line 259, we hope this highlights the main point of this figure, to show that 95% of the 1000 bootstrap comparisons between the control and e.g. the 10% change to `effgw_oro`,**

have fewer than 13, 19, and 20 of 117 output fields rejected for K-S, C-VM, and M-W tests respectively, and 1 2, and 2 of 117 output fields rejected for BH-FDR corrected versions of each. The 95% level is chosen here as it is the inverse of our significance level $\alpha = 0.05$, and as such one could expect to see 5% of bootstrap iterations failing the test on random chance, thus if fewer than 5% have passed, this is a strong indicator that the two ensembles are significantly different. We use the percentiles here and throughout the manuscript as they are robust and resistant measures of the data when compared to mean / standard deviation [Wilks, 2019].

- *I find it interesting and might be worth discussing, in regard to Figure 5, that there doesn't seem to be any correlation of between the false positives for MVK and BH-FDR approaches. On the one test day that fails BH-FDR the MVK approach is nowhere near failing.*
  Text has been added to the operation results section as it seems to be caused by the MVK approach not taking into account the value of p-values, only that there are enough below the threshold, while the BH-FDR approach yielded a failing overall result when one particular p-values was very small.

**2.2 Reviewer 2**

- *While the adoption of the MVK test is motivated by previous studies cited in this manuscript, there are alternatives to the MVK test that could also be considered before applying an FDR correction. For instance, the Anderson-Darling or Cramér-von Mises tests have advantages over the Kolmogorov-Smirnov test, such as being more sensitive to repeated deviations from the empirical distribution functions, detecting deviations in the tails of the distributions, and effectively handling skewed distributions.*
  The Cramér-von Mises and Mann-Whitney U test were added to the suite of tests, both are more sensitive to the tails, as mentioned Gentle [2003], and are still relatively quick to compute. It is now mentioned in the text, but they also exhibit greater statistical power in these results (as well as in theory), and the false positive rates are nearly similar to that of the K-S test.

- *As I understand it, the MVK test is performed on two independent ensembles with N=30 members, comparing the distribution of annual global means for 120 output fields (local null hypothesis). What is the rationale for choosing N=30 specifically, rather than other values of N? The reported results may vary with different values of N. In addition to the caveat of using a "low ultra-resolution model," the reported statistical results are conditional on a very specific value of N. It would be important to explore lower and higher N values to provide evidence of the effect or perhaps non-effect of the FDR correction under different sample size scenarios.*
  The $N = 30$ choice was based on earlier results from Mahajan [2021] where an ensemble of N=30 was found to balance statistical power and computational time. We have now conducted a similar analysis here which is described in Section 4.6 and visualized in Fig. 8. Similar to prior work, we also find that larger ensembles yield stronger statistical power. We also reach the same conclusion that an increase in statistical power by increasing ensemble size is associated with an increase in computational cost and N=30 strikes a good balance between the two. Developers always have the option to increase the ensemble size as needed.

- *Furthermore, why are "annual global means" the only statistic of interest rather than regional means or perhaps extreme value statistics, such as the maximum or minimum values over the same time-period? Testing for equality of distributions of global means represents just one focused statistical aspect of a set of climate ensembles with hundreds of output fields. I believe that many other tests beyond those proposed in this paper could be considered and of interest,*

*potentially leading to different statistical assessments of the ensembles and a wide range of results. Even under a coarse spatial resolution, it is to be expected that tests of a local mean or some other local statistic may lead to a different assessment than those resulting from a global statistic. This raises the question of what does it mean to study and test "climate reproducibility"? That variations of the model simulations provide similar global means?*

**We appreciate this comment and agree that regional means, local diagnostics, and extremes can provide valuable perspectives on ensemble behavior. In this work, however, we focus on annual global means, following previous work [Baker et al., 2015, Mahajan et al., 2017]. Atmospheric variability is highly heterogeneous in space and time, with sharp gradients and short-term fluctuations that complicate regional analyses. Global averaging suppresses this weather-driven noise and reduces dimensionality, making it more suitable for applying the ensemble consistency framework. For the ocean model component, where the ocean is more homogeneous we have implemented a test that accounts for spatial variability as well conducting tests at each grid point [Mahajan, 2021]. Baker et al. [2016] also use a similar approach and note that while global averaging can obscure signals in the ocean, it remains effective and appropriate for atmospheric applications. Further, extremes require much larger ensembles to be reliably assessed. Mahajan et al. [2017] demonstrated that with ensembles of about sixty members, extremes of temperature and precipitation were often statistically indistinguishable even when mean distributions diverged, underscoring their limited sensitivity in this context.**

- *The false positive rates and power analysis presented in the manuscript consider a one-at-a-time variation of the tuning parameters. Should simultaneous variations be considered to study interactions between the "effgw_oro" and "clubb-c1" parameters and understand how these interactions impact the rates and power? I realize that considering all possible interactions may create a large computational burden. However, selecting a few potential interactions of interest could provide further evidence in the application of an FDR correction to assess the reproducibility of climate simulations.*

  **We have considered just two tuning parameters here out of hundreds of available adjustments. Varying the two together may indeed yield interesting results, and could be indicative of how the test responds to a variety of changes. The purpose of the power analysis by gradually modulating tuning parameters in a controlled manner is also to inform the developers assessing the test results on the sensitivity of the test [Mahajan et al., 2017, 2019] Combining changes to both parameters takes away from this simpler approach, since tuning parameter changes can have non-linear effects which may be hard to disentangle. We have, however, added a set of ensembles based on varying a third tuning parameter, which can be seen throughout the results section (sec. 4). Additionally, using the global mean as the test statistic of interest makes the output of this test easier to understand at a glance by model developers in an operational setting and if further understanding is needed then additional tests can be performed.**

**2.3   Reviewer 3**

- *Limited Generalizability: The study's findings are highly specific to:*
  - *A single model (E3SM v2.1) at ultra-low resolution ( 7.5° atmosphere)*
  - *Annual global means of standard output variables*
  - *A particular computational environment (Argonne's Chrysalis machine with Intel compiler)*

*The authors acknowledge some limitations but do not adequately demonstrate that their approach generalizes beyond this narrow context. Testing at production resolutions, with other models, or with different climate statistics (regional patterns, extremes, seasonal cycles) is essential to support claims of broader applicability.*

**We conducted additional simulations at NE30 ($1°$) horizontal resolution to address the applicability of these results to production resolutions (one control, and one with an increased `clubb_c1` parameter). These two ensembles, and the additional `zmconv_c0_ocn` set of ensembles were all performed on the USAF HPC11, using the GNU compiler, further adding to the broader applicability. Testing this method in other Earth system models is beyond the scope of this manuscript and as mentioned above, global means are used to simplify and make the results accessible to model developers. Additionally, we are currently in the process of applying these methods to newer versions of E3SM, which include newer programmatic paradigms and algorithms for sub-grid scale process representations. However, we believe that the testing infrastructure is well designed to be applied to other models. In the future, we plan to seek out collaborations with other modeling communities to assess the generalizability of our testing methodology.**

- *Statistical Assumptions and Dependencies: The paper inadequately addresses:*
  - *The correlation structure among the 117 tested variables, which violates the independence assumption of standard BH-FDR*
  - *Whether the "theoretical" critical value of 1 for rejecting the global null hypothesis holds beyond their specific ensemble configuration*
  - *The choice of $q^* = \alpha = 0.05$ for FDR control without justification*

**It is an excellent point that the statistical dependence of the 117 output fields may violate the original assumptions of the BH-FDR correction. The BH-FDR correction is applied to the p-values resulting from the comparison of the global annual mean of each field from two 30 member ensembles, some of which are correlated positively, some are correlated negatively, and many are uncorrelated. The critical value of 1 is supported by the original methodology in Benjamini and Hochberg [1995], and holds for other cases, such as the results in Ventura et al. [2004]. Justification was added for using $q^* = \alpha = 0.05$ in section 2.3.**

- *Incomplete Methodological Comparisons: The manuscript lacks crucial comparisons with alternative approaches:*
  - *Permutation testing: While the authors mention that MVK "was found to give similar results as compared to permutation testing" (citing Mahajan et al., 2019b), they do not actually compare BH-FDR against permutation-based methods for their specific application.*
  - *Alternative multiple testing corrections: No comparison with other FDR methods (e.g., Benjamini-Yekutieli for dependent tests) or other approaches like the False Discovery Exceedance method.*

**Additional statistical tests were performed, as mentioned in our responses to other reviewers. We are using the same testing setup as described in [Mahajan et al., 2017], where the test statistic to evaluate the null hypothesis (that two simulation ensembles are statistically similar) is the number of variables rejecting a local null hypothesis. Here, we are adding the false discovery rate correction to it. So, our specific application has not essentially changed between then and now.**

They showed that permutation testing provided similar results as bootstrapping, when the critical threshold of the test statistic for the null hypothesis testing was computed via bootstrapping with a control ensemble or with permutation testing. Since, both are re-sampling approaches we believe they would continue to provide similar results. Also, the implementation of MVK in E3SM nightly suite is based on the bootstrapping version. We thus focus on comparisons of the BH-FDR approach against bootstrapping here. Text was added in section 2.3 to describe the previous examination of other correction methods, including Benjamini-Yekutieli and Bonferroni methods, which yielded lower power than both the MVK and BH-FDR approaches for this set of experiments.

These results match with those of Ventura et al. [2004], finding that the assumptions in Benjamini and Yekutieli [2001] were too restrictive, while Benjamini and Hochberg [1995] was simple to compute and held up for dependent data.

- *Limited Scope of Reproducibility Testing: Testing only annual global means represents a minimal assessment of climate reproducibility. Climate models are evaluated on their ability to simulate regional patterns, variability, extremes, and trends. A model could pass this test while having significant regional biases or incorrect variability patterns.*

  The global means as a test of climate reproducibility is intended a first-order test, as mentioned above in response to Anonymous Reviewer 2, local diagnostics, and extremes can provide valuable perspectives on ensemble behavior, however, in this work, we focus on annual global means, following previous work [Baker et al., 2015, Mahajan et al., 2017].

- **Minor Comments**

  - *The power analysis covers only two parameters. Testing additional parameters with different sensitivities would strengthen the conclusions.*
    An additional tuning parameter from a third physical parameterization scheme was added to the two parameters examined, with broadly similar results.

  - *The operational validation period (53 days) is quite short for drawing robust conclusions about false positive rates.*
    This is true, the operational period was chosen as it was a period where the model code base passed bit-for-bit testing for this configuration, and could not be expanded. It is intended as a real-world operational example where code changes were applied rather than a controlled experiment where tuning parameters were varied. Further examination of the one global null hypothesis rejection was added to the text in section 4.7.

  - *The manuscript reads more as a technical note about an incremental improvement to E3SM's testing infrastructure rather than a methodological advance in climate model evaluation*
    This point is well taken, we hope additional tuning parameters, simulation resolutions, statistical tests, computing platforms, and compilers highlight the broader applicability of the test as it is used.

**2.4 Reviewer 4**

- *I believe the paper effectively demonstrated that the false negative rate is lower while the false positive rate is nearly unchanged for BH-FDR compared to MVK in this specific use case.*
  *My biggest issue with the manuscript is that it seems to largely be a method development paper for ensuring statistical stability of a numerical model with an Earth system model used as the exemplar. Even though this is for a special issue focused on ensemble design, I think the article need more of an explicit relation to Earth System Dynamics. So, while I believe the paper has*

*strong merits for publication, I am not sure if this is the correct journal. Perhaps a more statistics-oriented journal would be appropriate.*

**As highlighted in our response to Anonymous Reviewer 3, we feel the broader applicability of this manuscript is enhanced by the inclusion of different resolutions, tuning parameters, and statistical tests.**

- *It's not clear to me what Section 4.4 is adding to the paper. They aren't showing any meaningful difference in the climate produced by two alternative compilers, but as they rightfully point out this is likely influenced by the fact that they are running ultra-low resolution simulations. So, they can't really comment on the ultimate effect of these compilers on the mean state of the climate outside of this specific configuration, which feels like a weak result.*

  **More careful description of the reasoning behind this experiment has been added in section 4.4, repeated here. The "...two simulation ensembles ("opt-O1" and "fastmath") were performed to apply the uncorrected and BH-FDR approaches to evaluate sensitivity of model results to compiler optimization flags, which are involved in the optimization of the underlying mathematics rather than model tuning parameters designed to account for varying physical processes."**

- *Minor comments*

  **The text corrections have been applied to the updated manuscript.**

**2.5  Reviewer 5**

- *There were repeated comments regarding the potential limitations of a 120-member ensemble in this work suggesting that this may not be enough simulations to fully represent the internal variability of MVK. While I found the rest of the paper to be a compelling defense of the update, this comment stuck with me and introduced a level of uncertainty into the conclusions that I was uncomfortable with. I would like to see some justification or supplementary analysis that suggests that this ensemble is sufficient for these results or at least can give some estimated range of uncertainty.*

  **An Earth system model's internal variability is difficult to diagnose as is illustrated by the need to run thousands of years of earth system models with pre-industrial forcings conducted by each modeling center to quantify it and before arriving at initial conditions to begin historical simulations. We thus stated that a 120-member ensemble may only provide an estimate of model's internal variability, since an ensemble size in the order of thousands for each scenario would be computationally expensive even at ultra-low resolutions. Also, we are effectively using 2040 ensemble members in all our analysis (from 17 cases - control and parameter change simulations - of 120 ensemble-members each), for example to derive the critical value threshold for the tests (e.g. found to be 11 for MVK). The spread among the self-comparisons (control vs. control and others), where samples were drawn from a pool of only 120 members, is small (ranging from 10-12 for MVK). This suggests that a 120-member ensemble is a good representation of the internal variability of the system for the purposes of solution reproducibility testing. Also, our choice of the ensemble size of 120 was based on similar sizes used in previous studies [Baker et al., 2015, Mahajan et al., 2017]. Their tests have been under production for several years now and have been successful at identifying many bugs further suggesting that these ensemble sizes are likely sufficient for testing purposes..**

- *Can you include a statement in your paper or in the results section that describes the improvement in computational time with BH-FDR versus MVK? This would then defend the twofold*

*benefits of the BH-FDR that were described in the introduction of your paper.*
**The following text has been added to the conclusion and discussion section detailing this time improvement. "This computationally expensive threshold finding, where the ensembles represent ≈3700 node hours of computation time for ensemble generation, is demonstrated to be effectively eliminated, as the BH-FDR approach for all statistical tests can use a threshold of 1, while maintaining nearly the same, or improving statistical power."**

- *Minor comments*
    - Corrections and suggestions will be applied to the update manuscript
    - *Line 177: While this method intuitively seems robust, is there justification for the 1000 times sample?*
    The 1000 bootstrap iterations was somewhat arbitrary, but chosen as a large enough to have statistically significant results, based on earlier work in Mahajan et al. [2017], which used 500 iterations.
    - *On the topic of detectability and mainly a question of personal interest: based on the relative sensitivities, can you generalize the detectability of a parameter change based on its sensitivity?*
    **This is an interesting question, and we have not explored this possibility. Based on the two parameters examined here, they do seem closely related. An additional perturbed parameter ensemble, as mentioned in responses to other reviewers, may not offer as much additional insight into this, as its sensitivity in Qian et al. [2018] is in the middle, but it appears more sensitive to parameter changes in this work. More careful examination of this is left to future work.**

**2.6 Reviewer 6**

Here we will respond to specific comments.

- *The assumptions of the Benjamini–Hochberg False-Discovery-Rate should be explicitly stated and checked*
**Text has been added to section 2.3 relating to the applicability of the BH-FDR to the presented data.**

- *the work should be put in a more general context, including*
*https://gmd.copernicus.org/articles/15/3183/2022/*
*and the previous work*
*https://dl.acm.org/doi/pdf/10.1145/3468267.3470572*
*The former work could, in particular, help to comment on some choices made (e.g. only Kolmogorov-Smirnov and the choice of ultra-low resolution) even though the tests are different.*
**Adding the context of Zeman and Schär [2022] and Mahajan [2021], will enhance the applicability of this technique to additional testing frameworks, as our scope is currently limited to the E3SM Atmosphere Model.**

- *It seems that FDR is also used in one of the author's previous papers:*
*https://dl.acm.org/doi/pdf/10.1145/3468267.3470572*
*and it should be made clear how the current results differ from the one in the previous paper.*
**Further context will be added to the updated manuscript, but briefly, the FDR correction in Mahajan [2021] was applied to the p-values computed at each grid point in each 2- or 3-D output field, rather than the p-values of global means of the output fields, and was designed specifically for the MPAS-Ocean model.**

- *The case study is quite restricted. Possibly more cases could be considered. I also agree with the other reviewers that sometimes more explanation could be given on the choice of methods/parameters.*

  **As mentioned in responses to other reviewers, further parameters, resolutions, and statistical tests were examined to further the broader applicability of the manuscript.**

**2.7 Reviewer 7**

- *There is a lot of jargon in the paper in the introductory sections, which makes it harder to read for a non-statistician. Some of the definitions (like Type I errors) were not clear from the text (even though the text refers to Sec. 4 for Type I error definition, there is no further clarification in Sec. 4). I found some clarifications in Mahajan2017 and Mahajan2019, but I think it is ok to repeat definitions of basic concepts in this paper.*

  **More explanation has been added to contextualize the statistical themes and techniques used here in all sections (e.g. section 2.3).**

- *Equation 1 – maximum function has boolean input, instead of a number. Is there a typo? Also, why does p-level there depend in an index ("i") of a field to be tested? The authors list m=117 above, but in general m can be arbitrary large.*

  **This equation has been updated in the manuscript to more clearly reflect the algorithm used. Additionally a new figure 1 has been added which demonstrates the effect of this correction.**

- *It would be very interesting to see (if not in this publication, then maybe in future ones) which simulations in the new framework (and possibly the old one) are deemed climate-changing by looking at their plots (standard latlon mean fields, zonal means, etc.), and evaluate them by an eyeball norm. Would a human expert also consider these simulations "climate changing" or not? Something similar is presented in Mahajan2017, but not exactly. There is an issue that the current work was done with an ultralow res E3SM (ne4 for the atmosphere), and this configuration of E3SM might not have established climatologies.*

  **Figure 7 has been added, using the ne30 resolution data, which compares a field which in the global mean sense is statistically different between our two test ensembles at this higher resolution as a demonstration, along with figure 8, which shows the cumulative distribution functions from the two ensembles for this output field.**

**References**

A. Baker, D. Hammerling, M. Levy, H. Xu, J. Dennis, B. Eaton, J. Edwards, C. Hannay, S. Mickelson, R. Neale, et al. A new ensemble-based consistency test for the community earth system model (pyCECT v1. 0). *Geoscientific Model Development*, 8(9):2829–2840, 2015. doi: 10.5194/gmd-8-2829-2015.

A. H. Baker, Y. Hu, D. M. Hammerling, Y.-H. Tseng, H. Xu, X. Huang, F. O. Bryan, and G. Yang. Evaluating statistical consistency in the ocean model component of the community earth system model (pyCECT v2.0). *Geoscientific Model Development*, 9(7):2391–2406, 2016. doi: 10.5194/gmd-9-2391-2016.

Y. Benjamini and Y. Hochberg. Controlling the false discovery rate: A practical and powerful approach to multiple testing. *Journal of the Royal Statistical Society: Series B (Methodological)*, 57(1):289–300, 1995. doi: 10.1111/j.2517-6161.1995.tb02031.x.

Y. Benjamini and D. Yekutieli. The control of the false discovery rate in multiple testing under dependency. *The Annals of Statistics*, 29(4):1165–1188, 2001. ISSN 00905364, 21688966. URL http://www.jstor.org/stable/2674075.

J. E. Gentle. *Random Number Generation and Monte Carlo Methods*. Springer-Verlag, 2003. ISBN 0387001786. doi: 10.1007/b97336. URL http://dx.doi.org/10.1007/b97336.

S. Mahajan. Ensuring statistical reproducibility of ocean model simulations in the age of hybrid computing. In *Proceedings of the Platform for Advanced Scientific Computing Conference*, PASC '21, New York, NY, USA, 2021. Association for Computing Machinery. ISBN 9781450385633. doi: 10.1145/3468267.3470572.

S. Mahajan, A. L. Gaddis, K. J. Evans, and M. R. Norman. Exploring an ensemble-based approach to atmospheric climate modeling and testing at scale. *Procedia Computer Science*, 108:735–744, 2017. ISSN 1877-0509. doi: 10.1016/j.procs.2017.05.259. International Conference on Computational Science, ICCS 2017, 12-14 June 2017, Zurich, Switzerland.

S. Mahajan, K. J. Evans, J. H. Kennedy, M. Xu, M. R. Norman, and M. L. Branstetter. Ongoing solution reproducibility of earth system models as they progress toward exascale computing. *The International Journal of High Performance Computing Applications*, 33(5):784–790, 2019. doi: 10.1177/1094342019837341.

Y. Qian, H. Wan, B. Yang, J.-C. Golaz, B. Harrop, Z. Hou, V. E. Larson, L. R. Leung, G. Lin, W. Lin, P.-L. Ma, H.-Y. Ma, P. Rasch, B. Singh, H. Wang, S. Xie, and K. Zhang. Parametric Sensitivity and Uncertainty Quantification in the Version 1 of E3SM Atmosphere Model Based on Short Perturbed Parameter Ensemble Simulations. *Journal of Geophysical Research: Atmospheres*, 123 (23):13,046–13,073, 2018. doi: 10.1029/2018JD028927.

V. Ventura, C. J. Paciorek, and J. S. Risbey. Controlling the proportion of falsely rejected hypotheses when conducting multiple tests with climatological data. *Journal of Climate*, 17(22):4343 – 4356, 2004. doi: 10.1175/3199.1.

D. S. Wilks. Chapter 3 - empirical distributions and exploratory data analysis. In D. S. Wilks, editor, *Statistical Methods in the Atmospheric Sciences (Fourth Edition)*, pages 23–75. Elsevier, fourth edition edition, 2019. ISBN 978-0-12-815823-4. doi: 10.1016/B978-0-12-815823-4.00003-1.

C. Zeman and C. Schär. An ensemble-based statistical methodology to detect differences in weather and climate model executables. *Geoscientific Model Development*, 15(8):3183–3203, 2022. doi: 10.5194/gmd-15-3183-2022.